## RESEARCH ARTICLE

# Effects of microtubule (de)tyrosination on the morphology and motility of *Trypanosoma brucei* and cross-talk with polyglutamylation

Marinus Thein[1], Hannes Wunderlich[2], Lucas Brehm[1,*], Stella Wagner[1], Matthias Weiss[2,‡] and Klaus Ersfeld[1,‡]

## ABSTRACT

Post-translational modifications (PTMs) of microtubules control many aspects of their functionality. Specifically, the C-terminal tails of α- and β-tubulin harbour a complex array of PTMs, including polyglutamylation and the reversible detyrosination/tyrosination. The spatial proximity of these two distinct sets of PTMs suggests the possibility of a functional cross-talk between polyglutamylation and (de)tyrosination. In this study, we employ gene deletion and overexpression of the enzymes tubulin-tyrosine ligase (TTL) and tubulin-tyrosine carboxypeptidase (VASH) to provide a detailed analysis of the effects of (de)tyrosination on the protozoan parasite *Trypanosoma brucei*. While the deletion of either of the enzymes is not lethal, cells exhibit subtle morphological defects, resulting from both hyper- and hypotyrosination. Additionally, hypertyrosination leads to defects in motility, characterised by an increase in tumbling motion. Using the TTL and VASH deletion cells in conjunction with our previously generated trypanosomes deficient in polyglutamylation, we uncovered a cross-talk between these two PTMs. The process of microtubule detyrosination enhances polyglutamylation, which, in turn, stimulates efficient detyrosination, thus establishing a positive feedback loop between these two PTMs.

KEY WORDS: *Trypanosoma brucei*, Cytoskeleton, Microtubules, Post-translational modifications

## INTRODUCTION

Microtubules represent a fundamental component of the eukaryotic cytoskeleton. The basic structural element of microtubule filaments is constituted by α- and β-tubulin proteins, which exhibit a high degree of conservation in their sequence across diverse organisms (Little and Seehaus, 1988; Nogales, 2015). However, microtubules fulfil a wide variety of functions. They are involved in most intracellular transport processes and are essential for motility of many cell types by forming the axoneme of flagella and cilia. Depending on their specific functions, the biophysical properties of microtubules vary even within a single cell, ranging from filaments displaying a high degree of dynamic instability to very stable microtubule arrays (Gull, 1999; Mitchison and Kirschner, 1984). In many protozoan cells, possibly in place of intermediate filaments, which are absent in protozoa, microtubules also have important structural functions (Ferreira and Frischknecht, 2023). Examples of such structures include the ventral disc of *Giardia* ssp., the apical complex of *Plasmodium* ssp. and, within the class of kinetoplasts, the formation of a subpellicular cytoskeleton (Elmendorf et al., 2003; Morrissette and Sibley, 2002; Sinclair and de Graffenried, 2019). The subpellicular cytoskeleton is a highly ordered nematic array of microtubule filaments that defines the spindle-like shape of these cells (Hemphill et al., 1991; Sherwin et al., 1987). Moreover, this array is extremely stable and does not exhibit the dynamic instability characteristic of cytoplasmic microtubule arrays in mammalian cells (Sinclair and de Graffenried, 2019). Given the high degree of conservation of the overall sequence and structure of α- and β-tubulin and of the resulting microtubules throughout the domain of eukaryotes, this variety in physical properties and functions is not merely an intrinsic factor of α- and β-tubulin proteins. Rather, regulation occurs at multiple levels, eventually securing a desired activity and function. Many organisms express several isotypes of α- and β-tubulins and/or a wide range of microtubule-associated proteins that modulate microtubule functionality (Bodakuntla et al., 2019). Moreover, the presence of a variety of post-translational modifications (PTMs) of α- and β-tubulin provides an additional regulatory layer (Janke and Bulinski, 2011; MacTaggart and Kashina, 2021). Collectively, the combination of these regulatory levels is referred to as the 'tubulin code' (Janke and Magiera, 2020; Verhey and Gaertig, 2007).

PTMs of the tubulin proteins have been recognised for decades (Barra et al., 1973; Edde et al., 1990; L'Hernault and Rosenbaum, 1985; Raybin and Flavin, 1977; Redeker et al., 1994); however, until recently, comparatively little research had been conducted on the detailed functions of these modifications in the regulation of microtubules. The first tubulin PTM to be described was the unusual detyrosination/tyrosination cycle (Barra et al., 1973). Here, the C-terminal tyrosine of α-tubulin is cleaved off by specific carboxypeptidases. The re-ligation of tyrosine to α-tubulin is then catalysed by a single tubulin-tyrosine ligase (TTL), which was first purified from porcine brain by Murofushi in 1980 (Murofushi, 1980). The removal of tyrosine from tubulin is more complex, as in mammals three different enzymes are able to catalyse this reaction, Vasohibins 1 and 2 (also known as VASH 1 and 2), and Microtubule-Associated-Tyrosine-Carboxy-Peptidase (MATCAP) (Aillaud et al., 2017; Landskron et al., 2022; Nieuwenhuis et al., 2017).

[1]Molecular Parasitology, Department of Biology, University of Bayreuth, Universitätsstraße 30, 95447 Bayreuth, Germany. [2]Experimental Physics I, Department of Physics, University of Bayreuth, Universitätsstraße 30, 95447 Bayreuth, Germany.
*Present address: Institute of Genetics, Department of Biology, University of Erlangen-Nürnberg, 91058 Erlangen, Germany.

‡Authors for correspondence (klaus.ersfeld@uni-bayreuth.de; matthias.weiss@uni-bayreuth.de)

M.T., 0009-0006-2179-4710; H.W., 0009-0000-2262-1412; L.B., 0009-0006-8006-7919; S.W., 0009-0002-6807-8624; M.W., 0000-0001-8814-9915; K.E., 0000-0002-9995-8010

Furthermore, the VASH enzymes require a cofactor (SVBP) for activity (Aillaud et al., 2017). The function of this reversible detyrosination/tyrosination cycle has long been enigmatic, and depletion of TTL in cultured cells showed only minor phenotypes (Webster et al., 1990). However, subsequent research demonstrated that transgenic mice lacking TTL died within 24 h after birth (Erck et al., 2005). This perinatal lethal phenotype was associated with defects in neuronal organisation, and further studies revealed brain abnormalities in *Ttl* knockout mice. Later, it was shown that impaired tyrosination affects the association of motor proteins and CAP-Gly-type microtubule-end binding proteins with microtubules, thus offering a potential molecular basis for the observed organismal phenotypes (Pagnamenta et al., 2019; Peris et al., 2022, 2006).

Following the cloning of the porcine *TTL* and the availability of reference genomes, it was revealed that the core catalytic domain of TTL defined a large family of homologues enzymes (Ersfeld et al., 1993; Janke et al., 2005). They were identified as either tubulin-polyglutamylases or tubulin-polyglycylases (tubulin-tyrosine-ligase like; TTLLs) (Janke et al., 2005; Rogowski et al., 2009; van Dijk et al., 2007). These enzymes catalyse either the addition of polyglutamate or polyglycine side chains to specific glutamic acid residues with the C-terminal fifteen amino acids of both α- and β-tubulin. As is the case with the detyrosination/tyrosination cycle, these PTMs have been shown to be conserved during evolution (Bre et al., 1996; Rogowski et al., 2009; Schneider et al., 1997; Weber et al., 1996). Whereas the function of polyglycylation is poorly understood, polyglutamylation deficiencies in genetic knockout or knockdown models frequently manifest severe phenotypes associated with stable microtubule structures, including disorganised axonemes within cilia and flagella. A number of ciliopathies, such as infertility due to immotile sperm cells, are now associated with mutations in TTLL enzymes (Bedoni et al., 2016; Giordano et al., 2019; Van De Weghe et al., 2022).

Significant advances in our understanding of the functions of microtubule PTMs have been achieved through the study of single-celled organisms, including *Chlamydomonas reinhardtii* and *Tetrahymena thermophila* (Chhatre et al., 2025; Gaertig, 2000; Kubo and Oda, 2019; Suryavanshi et al., 2010; Xia et al., 2000). They possess well-characterised microtubule structures, and they can be readily genetically manipulated. In the present study, the unicellular parasite *Trypanosoma brucei* is employed as a model organism to study aspects of the tubulin code. *T. brucei* is a useful model organism as, in addition to the availability of a range of techniques to manipulate this organism, the biology of microtubule PTMs is less complex than in most other organisms (Sinclair and de Graffenried, 2019). Trypanosomes do not express variant isotypes of α- and β-tubulins, polyglycylation as a PTM is absent, and the detyrosination reaction is catalysed by only one enzyme (TbVASH), in contrast to the three in mammalian cells (van der Laan et al., 2019). It is noteworthy that, in *T. brucei*, the detyrosination/tyrosination cycle impacts both α- and β-tubulin, since β-tubulin also terminates with a tyrosine (Schneider et al., 1997; van der Laan et al., 2019).

In addressing the function of the various microtubule PTMs, a question that has so far received little attention is the possibility of cooperativity between PTMs. This is relevant given that the 'tubulin code' is evidently defined as the set of all possible combinations of PTMs (Janke and Magiera, 2020). Moreover, from a mechanistic perspective, a certain degree of cross-talk between those specific PTMs is a reasonable hypothesis because at least polyglutamylation, polyglycylation and the detyrosination/tyrosination cycle occur in close proximity at the C-terminal tail of the tubulin proteins (Roll-Mecak, 2020). In two recent studies, *in vitro* assays using synthetic TTLL substrate peptides or semisynthetic tubulin revealed that detyrosinated α-tubulin tail peptides are a much better substrate for the polyglutamylase TTLL6 than tyrosinated peptides and that, vice versa, polyglutamylation promotes detyrosination (Ebberink et al., 2023; Mahalingan et al., 2020). Based on our functional characterisation of several tubulin polyglutamylases in *T. brucei*, we are now in a position to address possible dependencies between microtubule PTMs *in vivo* (Jentzsch et al., 2020, 2024).

In the present study, an in-depth analysis of the detyrosination/tyrosination cycle in *T. brucei* was conducted, resulting in the discovery of motility alterations in TbVash knockout cells in comparison to wild-type cells. In addition, the potential for cross-talk between the detyrosination/tyrosination cycle and polyglutamylation was examined. Our data revealed the existence of a positive feedback loop, whereby an increase in detyrosination serves to reinforce polyglutamylation and an increase in polyglutamylation, in turn, reinforces detyrosination.

## RESULTS
### Effects of gene deletion of *VASH* and *TTL* on the microtubule cytoskeleton and cell dynamics

The simplified cycle of tubulin detyrosination/tyrosination in *T. brucei*, catalysed by the only carboxypeptidase VASH and the tyrosine ligase TTL, respectively, allows the interrogation of the significance of this microtubule PTM by gene deletion of the respective genes. We successfully generated gene knockouts of the *VASH* and *TTL* gene ($vash^{-/-}$ and $ttl^{-/-}$), respectively, by replacing the open reading frames with antibiotic resistance genes (Fig. S1A). A PCR analysis using genomic DNA confirmed the correct insertion of the resistance cassettes, while an additional quantitative PCR (qPCR) validated the absence of transcripts (Fig. S1B,C). Additionally, we constructed a rescue cell line for the $vash^{-/-}$ cell line ($vash^{-/-}$-Rescue), ectopically expressing a myc-fused *vash* gene at slightly elevated levels, compared to the wild type, and a *vash*-overexpressing cell line based on the $ttl^{-/-}$ cell line ($ttl^{-/-}$ - $Vash^{OE}$) (Fig. S1C,D).

First, we analysed the levels of microtubule tyrosination by western blotting of cytoskeletal preparations. We used the YL1/2 antibody, which is specific for tyrosinated α-tubulin and the TAT antibody, which binds α-tubulin independent of tyrosination levels, as a loading control (Fig. 1A) (Kilmartin et al., 1982; Woods et al., 1989). The $vash^{-/-}$ cell line showed a significant increase in tyrosinated α-tubulin compared to the wild-type 427 cells. In the $vash^{-/-}$ - *Rescue* cell line, α-tubulin tyrosination was reverted to wild-type level. In contrast, $ttl^{-/-}$ cells showed a decrease in tyrosination, which was further enhanced in $ttl^{-/-}$- $Vash^{OE}$ cells (Fig. 1A). Conversely, probing blots with antibodies against detyrosinated α- and β-tubulin showed complete absence of detyrosination in $vash^{-/-}$ cells, confirming that VASH is the only carboxypeptidase involved, acting both on α- and β-tubulin (Fig. 1B). Using immunofluorescence, we analysed the distribution of tyrosinated microtubules in these cell lines. In the $vash^{-/-}$ cell line, the entire cytoskeleton is tyrosinated and reverts to its wild-type appearance in the rescue cell line (Fig. 1C; Fig. S2A). The $ttl^{-/-}$ cell line showed a decrease in tyrosination compared to that in wild-type cells, with some faint signal remaining at the posterior cell poles. This remnant signal was completely abolished in the $ttl^{-/-}$ - $Vash^{OE}$ cells. Interestingly, the strong staining of the basal body with YL1/2 remained unaffected in all cell lines (Fig. 1C; Fig. S2A). However, the YL1/2-positive staining of the basal body is not necessarily due to the presence of tyrosinated tubulin, because it has been shown that a different component of the basal body, the transition fibre protein RP2, cross-reacts with this antibody (Andre et al., 2014). The analysis of isolated flagella showed extensive microtubule tyrosination in the $vash^{-/-}$ cell line (Fig. 1D).

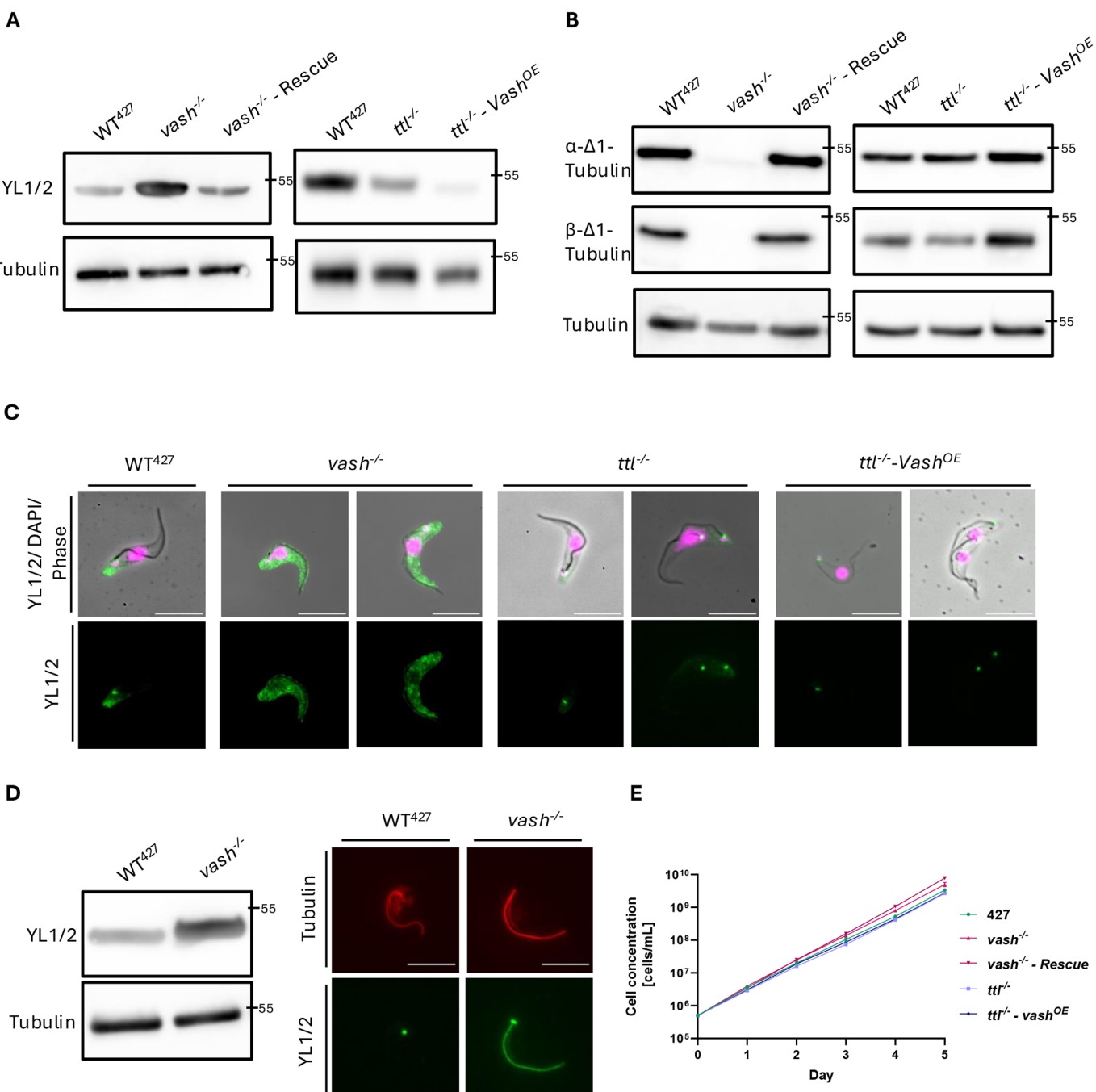

**Fig. 1. Knockouts of *TTL* and *VASH* reversibly impact the tyrosination levels.** (A) Western blot analysis of cytoskeletal fractions of wild-type (WT[427]), *vash*[−/−], *vash*[−/−] - *Rescue*, *ttl*[−/−] and *ttl*[−/−] - *Vash*[OE] cells. The antibody YL1/2 recognises tyrosinated α-tubulin. The anti-tubulin antibody signal serves as a loading control. (B) Western blot analysis of cytoskeletal fractions of WT[427], *vash*[−/−], *vash*[−/−] - *Rescue*, *ttl*[−/−] and *ttl*[−/−] - *Vash*[OE] cells. The antibodies each recognise detyrosinated α- or β-tubulin (α-Δ1/ β-Δ1). The anti-tubulin antibody signal serves as a loading control. (C) Immunofluorescence analysis of tyrosination levels of cytoskeletons of WT[427], *vash*[−/−], *ttl*[−/−] and *ttl*[−/−] - *Vash*[OE] cells. The YL1/2 signal is depicted in green. The DNA was stained with DAPI (magenta). (D) Analysis of the levels of tyrosinated tubulin in isolated flagella of WT[427] and *vash*[−/−] cells through western blotting and immunofluorescence. The YL1/2 signal is depicted in green and the tubulin signal in red. (E) Cumulative growth curves of WT[427], *vash*[−/−], *vash*[−/−] - *Rescue*, *ttl*[−/−] and *ttl*[−/−] - *Vash*[OE] cells. Three biological replicates, measured three times each, were used. The *y*-axis is plotted logarithmically. Scale bars: 10 µm. The 55 kDa band of the used PageRuler Protein Ladder is indicated on all blots as a molecular weight orientation.

In wild-type cells, the axonemal microtubules of the flagella are known to be completely detyrosinated (Schneider et al., 1997). However, none of these fundamental changes of the microtubule tyrosination status significantly impaired growth, generation time, cell cycle distribution and mitotic progression of the various cell lines (Fig. 1E; Tables S1, S2).

Analysis of cell morphology in these cell lines revealed some minor but significant changes. The *vash*[−/−] cell line showed a small, but statistically significant, decrease in cell length. As the generation time of the population is the same as wild type, this increase is unlikely due to a delayed cell cycle progression (Table S1). This phenotype could be rescued by re-expression of the *VASH* gene

(Fig. 2A). Flagellar length in *vash*⁻/⁻ cells was not significantly different from that in wild-type cells.

The *ttl*⁻/⁻ cell line was almost indistinguishable from wild type. Occasionally, elongated cells were observed, but they were not representative of the population (Fig. 2B). However, in the *ttl*⁻/⁻ - *Vash*$^{OE}$ cell line, with almost complete absence of tyrosination, distinct morphological defects were observed. These cells were shorter than wild-type cells, and the width of the posterior pole decreased significantly (Fig. 2B,C; Fig. S3A). Overall, these cells lost their typical spindle-like shape and instead appeared more bulbous

and shortened with a pointed posterior tip (Fig. 2C; Fig. S3B). In both knockout cell lines, a small percentage (<1%) of cytokinesis-deficient, multinucleated cells was found (Fig. S3C).

A previous study reported defects of the mitotic spindle and aberrant chromosome segregation after CRISPR-mediated *VASH* knockout, resulting in a severe growth phenotype (van der Laan et al., 2019). In contrast, we observed no significant change in cell cycle distribution using flow cytometry analysis (Fig. S4A). Likewise, the morphology of the mitotic spindle and the distribution of Kinesin 13-1, a key player during mitosis and necessary for correct

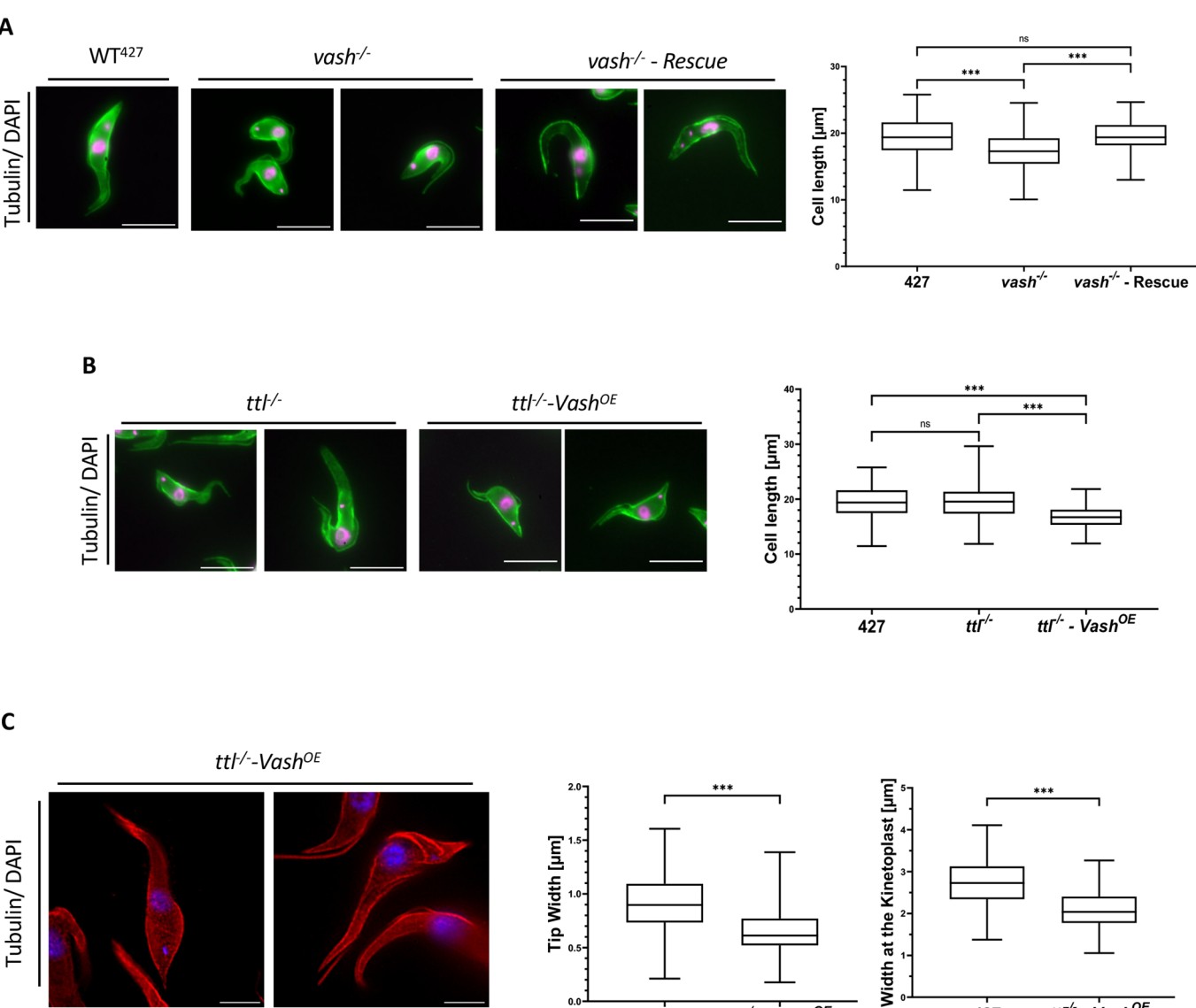

**Fig. 2. The (de)tyrosination has a small, but significant, impact on cellular morphology.** (A) Left: immunofluorescence of the morphology of WT$^{427}$, *vash*⁻/⁻ and *vash*⁻/⁻ - *Rescue* cells. The tubulin signal is depicted in green and the DAPI signal in magenta. Right: measured cell length of cells in G1-phase (one nucleus and one kinetoplast) of WT$^{427}$, *vash*⁻/⁻ and *vash*⁻/⁻ - *Rescue* cells. The boxes depict the cell length median and the interquartile range. The whiskers depict the range between the minimum and maximum values. Statistical significance is indicated above [Kruskal–Wallis test, not significant (ns)=P>0.05, ***0.0005>P, n=138 cells]. (B) Left: immunofluorescence of the morphology of *ttl*⁻/⁻ and *ttl*⁻/⁻ - *Vash*$^{OE}$ cells. The tubulin signal is depicted in green and the DAPI signal in magenta. Right: measured cell length of cells in G1-phase (one nucleus and one kinetoplast) of WT$^{427}$, *ttl*⁻/⁻ and *ttl*⁻/⁻ - *Vash*$^{OE}$ cells. The boxes depict the cell length median and the interquartile range. The whiskers depict the range between the minimum and maximum values. Statistical significance is indicated above (Kruskal–Wallis test, as in A, n=138 cells). (C) Left: expansion microscopy analysis of the morphology of *ttl*⁻/⁻ - *Vash*$^{OE}$ cells. The cells were stained with the anti-tubulin antibody (red) and the DNA was stained with DAPI (blue). Both the anti-tubulin and the DAPI signal were deconvoluted. Right: measured tip width and width at the kinetoplast of WT$^{427}$ *and ttl*⁻/⁻ - *VashExp* cells in G1-phase. The various lengths were measured as indicated in Fig. S3A. The boxes depict the cell length median and the interquartile range. The whiskers depict the range between the minimum and maximum values. Statistical significance is indicated above (Mann–Whitney-*U* test, ***0.0005>P, n=121 (tip width), 123 (width at the nucleus) and 138 (width at the kinetoplast) cells). Scale bars: 10 µm.

mitotic spindle assembly, was unchanged to the wild-type (Fig. S4B,C) (Chan et al., 2010). We used fluorescence *in situ* hybridisation analysis of mitotic spindle-mediated minichromosome segregation (Ersfeld and Gull, 1997) to assess possible mitotic segregation defects but observed no obvious deficiencies (Fig. S4D), a result congruent with the normal growth dynamics we observed at the population level.

Additionally, we analysed the impact of the microtubule tyrosination status on cell motility. To this end, we followed our previously established methodology based on trajectory analysis of individual cells (for details, see Jentzsch et al., 2024): trajectories of motile trypanosomes were automatically dissected into phases of run and tumble motion via the local straightness of the migration path. Here, cells were classified as being motile when their trajectory showed at least once a displacement of 20 μm (i.e. one cell length) within a period of 10 s; with that criterion, dead and/or immobilised cells were automatically discarded. For the remaining ensemble of trajectories, we determined the total fraction of positions that were assigned to the run phase ('run phase fraction'; $f_R$) and the probability density function (PDF) of the cells' velocity at these positions; $p(v_R)$. While $p(v_R)$ hardly varied between wild-type, *vash*$^{-/-}$, and *ttl*$^{-/-}$ -

*Vash*$^{OE}$ cells (Fig. 3A), the run phase fraction turned out to be about 30% larger for *vash*$^{-/-}$ ($f_R$=61%) than for wild-type ($f_R$=47%) and *ttl*$^{-/-}$ - *Vash*$^{OE}$ ($f_R$=47%) cells. Thus, *vash*$^{-/-}$ cells perform better in the motility assay. This is not due to increased swimming speed (Fig. 3B), but rather to cells being either shorter in the tumble phase or by being longer in the run phase. Both of these effects can increase the value of $f_R$. As a simple means to distinguish between these two possibilities, we inspected the periods spent in the run and tumble phases, $\tau_R$ and $\tau_T$, respectively. In particular, we determined all periods $\tau_R$ between a switch from tumble to run and back, i.e. we neglected run periods that included the first or last position of a trajectory as the cell had already switched to the run phase prior to recording the trajectory or it remained in the run phase even after stopping the recording. Accordingly, we determined values for $\tau_T$ as periods between switches from run to tumble and back. For consistency with our analysis on the cell velocities, in which velocity and trajectory straightness are determined on time scales of at least 1 s (for a detailed discussion, see Jentzsch et al., 2024), we discarded all periods $\tau_R$ and $\tau_T$ below 1 s. For the remaining periods, we inspected the cumulative distribution $P_{cum}(\tau)$, i.e. the integral of the PDF, since this quantity does not require the definition of a bin

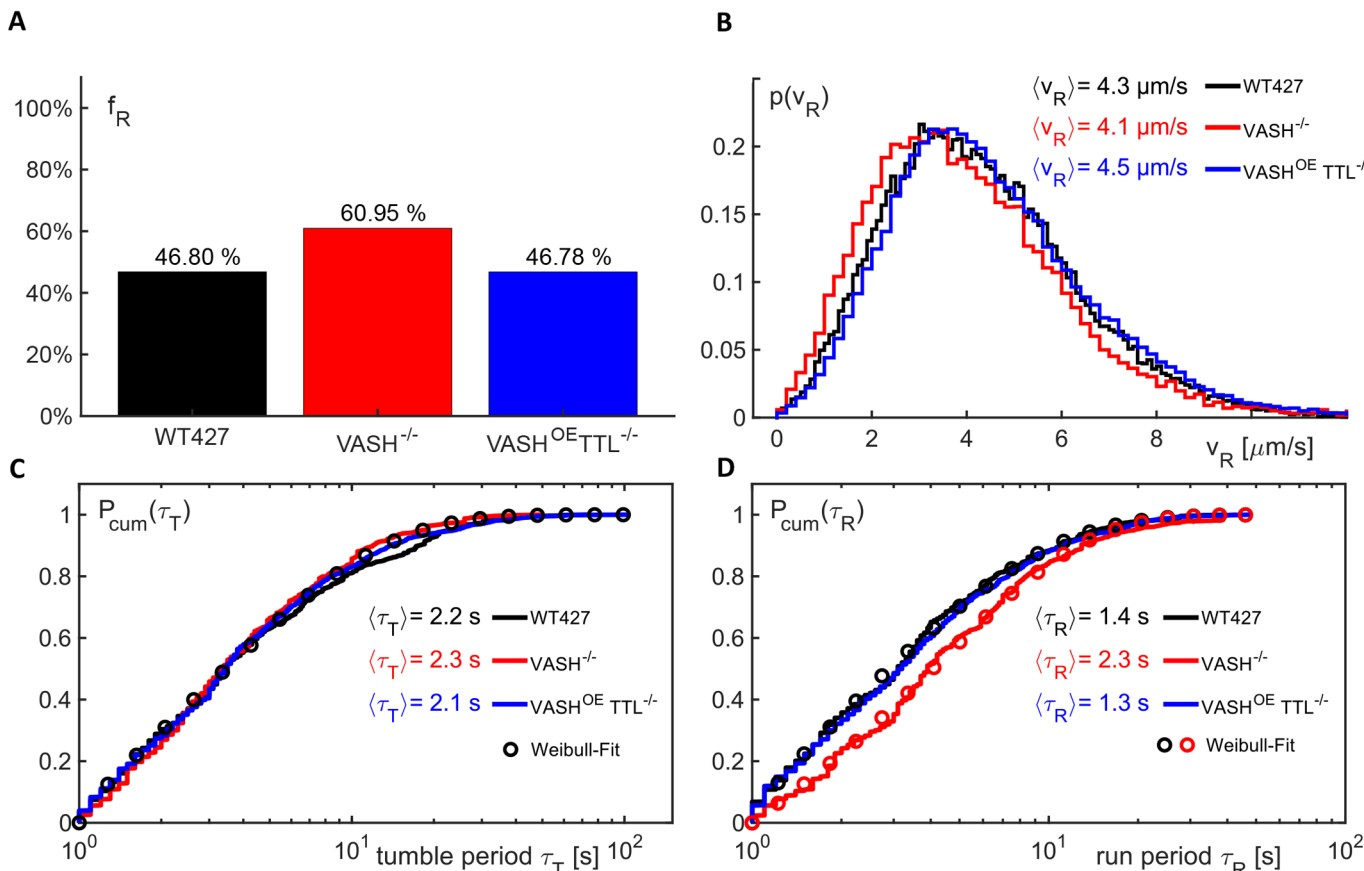

**Fig. 3. Microtubule hypertyrosination affects trypanosome run-tumble persistency.** (A) The fraction of positions in the run phase, $f_R$, obtained from large ensembles of motile cell trajectories that were automatically dissected into run and tumble phases (for technical details, see Jentzsch et al., 2024) revealed a marked increase for *vash*$^{-/-}$ cells compared to wild-type and *ttl*$^{-/-}$ - *Vash*$^{OE}$ cells. Hence, *vash*$^{-/-}$ cells remain longer or more often in the run state. (B) The probability distribution of cell velocities in the run phase, $p(v_R)$, is basically the same for all three cell lines, featuring very similar mean velocities. Cells of the *vash*$^{-/-}$ line show a tendency for slower velocities, i.e. the PDF is shifted slightly to lower values, but none of the cell line features a significantly higher or lower swimming speed. (C,D) The cumulative distribution of periods in the tumble phase, $P_{cum}(\tau_T; C)$, only shows little variations between the three cell lines and all are well captured by a Weibull distribution with a single parameter set (see main text for details). In contrast, the cumulative distribution of periods in the run phase, $P_{cum}(\tau_R; D)$, features a marked shift to larger periods for *vash*$^{-/-}$ cells, indicating that these cells remain for prolonged periods in the run state. Consequently, wild-type and *ttl*$^{-/-}$ - *Vash*$^{OE}$ cells are both captured by a Weibull distribution with a single parameter set, yet another parameter set is needed to capture the data for *vash*$^{-/-}$ cells (see main text for details). Please note the logarithmic *x*-axis.

width for a histogram. As a result, we observed that the cumulative distribution of tumble periods, $P_{cum}(\tau_T)$, was basically the same for all three cell lines, with mean/median values of 6.7 s/3.4 s (wild type), 5.5 s/3.3 s ($vash^{-/-}$), and 6.4 s/3.5 s ($ttl^{-/-} - Vash^{OE}$). The fairly different values for mean and median indicate already that tumble periods feature a heavy-tailed distribution, i.e. the mean is not a proper representative for the ensemble of values. In line with this notion, we found that a Weibull distribution with the general form $P_{cum}(\tau)=1-\exp\{-(\tau/\alpha)^\beta\}$ captured the data for all three cell lines when choosing the parameters as $\alpha=4$ s and $\beta=0.75$ (shown as an overlay in Fig. 3C). In contrast, the cumulative distribution of run periods, $P_{cum}(\tau_R)$, showed a marked shift to larger values for the $vash^{-/-}$ cell line (Fig. 3D). This visible shift is in agreement with a change in the mean/median values for the run period from 4.7 s/3.1 s (wild type) to 6.3 s/3.9 s ($vash^{-/-}$) and back to 5.0 s/3.2 s ($ttl^{-/-} - Vash^{OE}$). The roughly 30% increase in the mean/median run period from wild-type to $vash^{-/-}$ cells matches well the 30% increase in the run phase fraction for $vash^{-/-}$ ($f_R=61\%$ instead of $f_R=47\%$ in the wild type). We therefore conclude that $vash^{-/-}$ cells remain for prolonged periods in the run phase, rather than increasing the switching rate to the run state. Analogous to the statistics of tumble periods, the data for run periods are well described by a Weibull distribution, with parameters $\alpha=3.1$ s and $\beta=0.75$ for wild-type and $ttl^{-/-} - Vash^{OE}$ cells, but $\alpha=4.6$ s and $\beta=0.9$ for $vash^{-/-}$ cells (both shown as overlays in Fig. 3D). Thus, compared to those for wild-type cells, the run periods for $vash^{-/-}$ cells are prolonged, and their distribution becomes considerably less heavy-tailed, almost reaching a simple exponential distribution ($\beta=1$). The motility of $ttl^{-/-}$ cells was indistinguishable from that of wild-type cells.

Altogether, the analysis of the motility assay demonstrates, that the lack of microtubule detyrosination strongly impacts the likelihood of a cell being in either the running or the tumbling phase.

### Variations in tyrosination levels induce changes in polyglutamylation

Recent *in vitro* studies employing peptides or semisynthetic tubulin have suggested the potential for regulatory cross-talk between polyglutamylation and detyrosination. (Mahalingan et al., 2020; Ebberink et al., 2023). To address this possible cooperativity between various PTMs *in vivo*, we investigated the polyglutamylation levels in the $ttl^{-/-}$ and $vash^{-/-}$ knockout cell lines using antibodies recognising mono- and polyglutamylation. The $vash^{-/-}$ cells showed a strong decrease in their Poly-E and GT335 signal intensity, while the signal intensities in the $ttl^{-/-}$ cell line were unchanged (Fig. 4A; Fig. S5A). This correlation was also observed in isolated flagella (Fig. S5B). The Poly-E antibody recognises glutamyl side chains longer than three residues on both α- and β-tubulin, while the GT335 antibody binds to the branch point glutamyl side chains of both α- and β-tubulin and recognises chains of any length (Jentzsch et al., 2020, 2024; Rogowski et al., 2010; Shang et al., 2002; Wolff et al., 1992). Furthermore, we used the β-mono-E antibody (Bodakuntla et al., 2019), which recognises the specific polyglutamylation branching at E435 on β-tubulin (Fig. S5A). This antibody showed no change in its signal intensity in both knockout cell lines, compared to wild type, indicating that the formation of polyglutamylation branching points was unaffected by the high tyrosination levels (Fig. 4A; Fig. S5A). However, the elongation of glutamyl side chains was strongly reduced in the $vash^{-/-}$ cells. This phenotype was reverted in the $vash^{-/-}$ - *Rescue* cell line (Fig. 4A). To visualise these differences *in situ*, we mixed the wild-type population with the $vash^{-/-}$ population. We then double-labelled cells with Poly-E or GT335, and with the YL1/2 antibody. The latter strongly labels $vash^{-/-}$ cells (Fig. 4B; Fig. S5C). Congruent with the western blot analysis, cells with high tyrosination

levels, representing $vash^{-/-}$ cells, showed a strong signal decrease for both Poly-E and GT335 compared to the wild-type cells (Fig. 4B; Fig. S5C). In $vash^{-/-}$ - *Rescue* cells, Poly-E and GT335 signals reverted to wild-type levels (Fig. S5D).

### The impact of *VASH* deletion on polyglutamylation is non-uniform across the cell body

To resolve the spatial distribution of the inverse correlation between tyrosination and polyglutamylation, we analysed the subcellular localisation of the signal of Poly-E and GT335 in the wild-type, knockout and rescue cell lines (Fig. 4C). To quantify this, we measured the fluorescence intensity within a defined circular area of a diameter of 1 μm near the anterior and posterior poles. The resulting fluorescent ratio reveals the differences in signal intensity between the anterior and posterior pole of the cells ($I_A/I_P$). For both the GT335 and Poly-E we observed a significant $I_A/I_P$ decrease in the $vash^{-/-}$ cell line, while both the wild-type and $vash^{-/-}$ - *Rescue* cells showed similar ratios (Fig. 4C). This indicates that the polyglutamylation levels do not uniformly decrease throughout the cell body but mainly affect the anterior region of the cells. In general, microtubules at the posterior pole are harbouring newly incorporated tyrosinated tubulin and have yet to be detyrosinated by VASH. The lack of VASH in the knockout cell line, therefore, does not affect the natural tyrosinated state at this pole, while the anterior parts of the cells now remained tyrosinated. This change in distribution of the tyrosination pattern towards the anterior pole in return also reflects negatively on the proper formation of polyglutamylation.

Since high levels of tyrosination negatively correlate with polyglutamylation, we investigated how very low levels of tyrosinated tubulin influence polyglutamylation. The $ttl^{-/-}$ cell line showed no significant difference in polyglutamylation levels compared to the wild type. However, given that genetically encoded tyrosinated tubulin competes with detyrosination, we analysed our $ttl^{-/-}$ - $Vash^{OE}$ cell line, with further decreased tyrosination levels (Fig. 4D). Indeed, this led to an additional increase in polyglutamylation levels, most evident in the Poly-E signal (Fig. 4D). Overall, these findings provide *in vivo* evidence that microtubule tyrosination might have an inhibitory effect on polyglutamylation.

### Microtubule detyrosination and polyglutamylation regulate each other

Next, we asked whether polyglutamylation affect tyrosination. Therefore, we analysed the tyrosination levels in our previously established polyglutamylase knockout ($ttll1^{-/-}$) and RNA interference (RNAi) ($ttll12B$) cell lines (Jentzsch et al., 2020, 2024). The phenotypes of all cells where these genes were deleted or transcripts depleted showed defects in cytoskeletal organisation and a decrease in polyglutamylation levels, affecting both the presence of short and long glutamyl side chains (Jentzsch et al., 2020, 2024).

Analysing the $ttll1^{-/-}$ cell line, we found an increase in tyrosination levels by western blot analysis, confirming our previous data (Fig. 5A) (Jentzsch et al., 2024). We recently showed that the knockout of this polyglutamylase mostly affected small and long glutamyl side chain levels (Jentzsch et al., 2024). The branching point E435 on β-tubulin (using the β-mono-E antibody) is unaffected by the deletion of the *TTLL1* gene (Fig. 5A). This supports our previous classification of TTLL1 as an elongase (Jentzsch et al., 2024). Next, we analysed the $ttll12B$ - RNAi cell line, which was shown to have a decrease in long glutamyl side chains, also indicating role as elongase (Jentzsch et al., 2020). This cell line showed a strong increase in tyrosination levels upon induction of $ttll12B$ - RNAi, further suggesting a functional

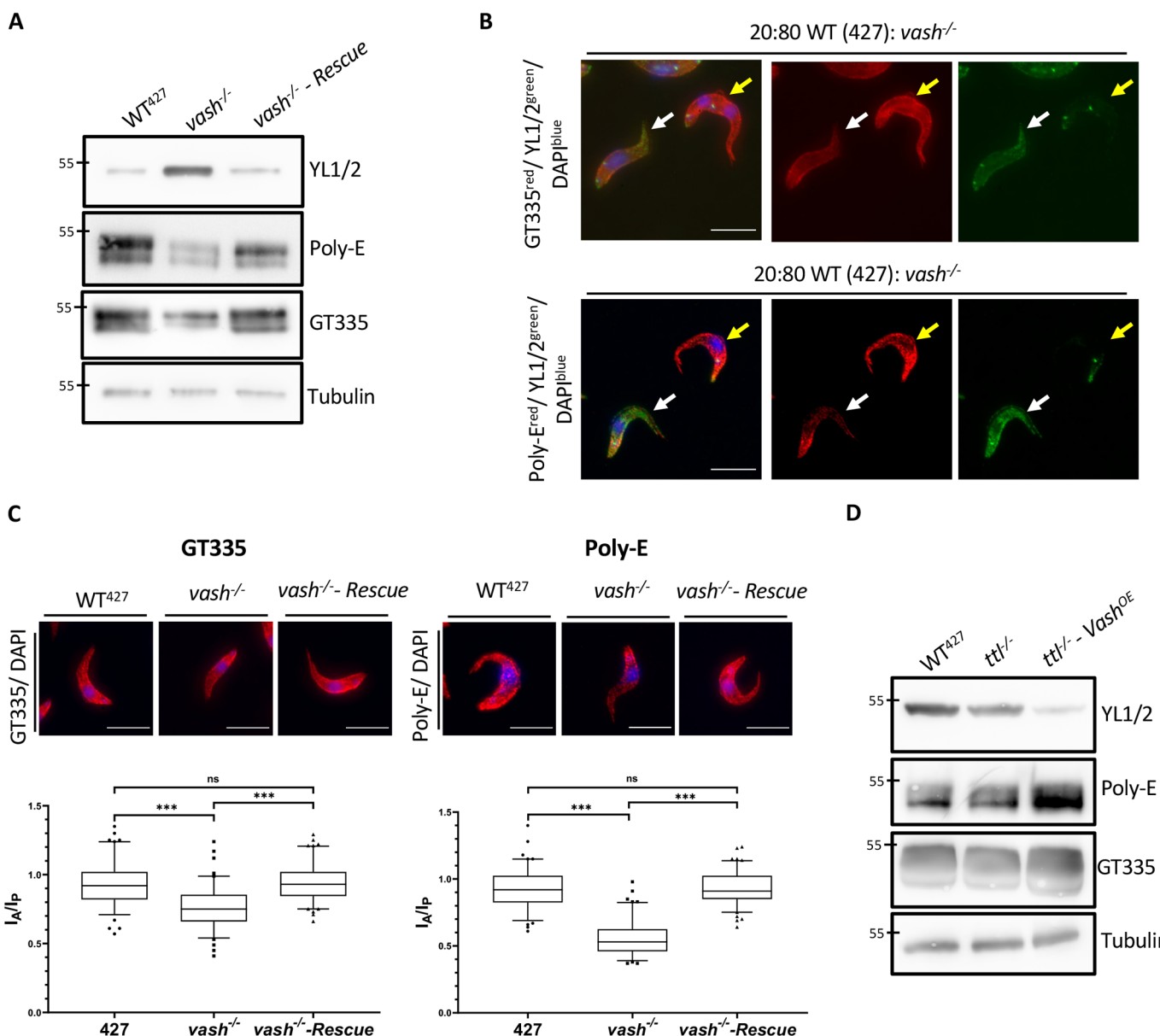

**Fig. 4. Tyrosinated tubulin levels inversely correlate with polyglutamylation.** (A) Western blot analysis of tyrosination and polyglutamylation levels in cytoskeletal extracts of WT[427], $vash^{-/-}$ and $vash^{-/-}$ - Rescue cells. The Poly-E antibody detects long glutamyl side chains on both α- and β-tubulin (Shang et al., 2002; Rogowski et al., 2010). The GT335 antibody recognises the branching points and small glutamyl side chains (Wolff et al., 1992; Rogowski et al., 2010) on both α- and β-tubulin. The poly-E and GT335 upper and lower bands correspond to α- and β-tubulin, respectively. (B) Immunofluorescence analysis of polyglutamylation levels of cytoskeletons of WT[427] (indicated by yellow arrows) and $vash^{-/-}$ (indicated by white arrows) cells mixed. Both cell lines were mixed in a 20:80 ratio prior to the preparation of the slides. The signal of GT335 or Poly-E is depicted in red, the signal of YL1/2 in green and the DAPI signal in blue. (C) Analysis of the signal distribution of GT335 and Poly-E in WT[427], $vash^{-/-}$ and $vash^{-/-}$ - Rescue cells. The signal intensity was measured in an area of equal size towards the anterior pole ($I_A$) and towards the posterior pole ($I_P$) and then divided by each other. The boxes depict the median and the interquartile range. The whiskers depict the range between the 5 and 95 percentile. Statistical significance is indicated above (Kruskal–Wallis test, ns=$P$>0.05, ***0.0005>$P$, $n$=101 cells). Representative images of the signal distribution in the three cell lines are depicted next to the plots, with the GT335 or Poly-E signal shown in red and the DAPI signal in blue. (D) Western blot analysis of tyrosination and polyglutamylation levels in cytoskeletal extracts of WT[427], $ttl^{-/-}$ and $ttl^{-/-}$ - $Vash^{OE}$ cells. The 55 kDa band of the used PageRuler Protein Ladder is indicated on all blots as a molecular weight orientation. Scale bars: 10 µm.

connection between polyglutamylation and detyrosination (Fig. 5B). Taken together, these data suggest that a decrease in polyglutamylation, and in particular long chain polyglutamylation, might reduce the capacity of microtubules to get detyrosinated by VASH, leading to an increase in tyrosination levels.

**Active Vash is required for PTM cross-talk**
To discriminate between the possibilities that the catalytic activity of VASH or the presence of VASH as an activator of polyglutamylases

is responsible for this correlation between detyrosination and polyglutamylation, we created a catalytically dead variant of Vash and expressed this mutant in the $vash^{-/-}$ cell line. We mutated the functionally important cysteine 94 of the catalytic triad to alanine, to create the dead variant, $vash^{-/-}$ - C94A (Fig. 5C; Fig. S1C) (Sanchez-Pulido and Ponting, 2016; Van Der Laan et al., 2019). Next, we analysed the tyrosination and polyglutamylation levels of this variant compared to the wild-type, knockout and functional rescue cell lines (Fig. 5D). The signal for tyrosinated tubulin in $vash^{-/-}$ and

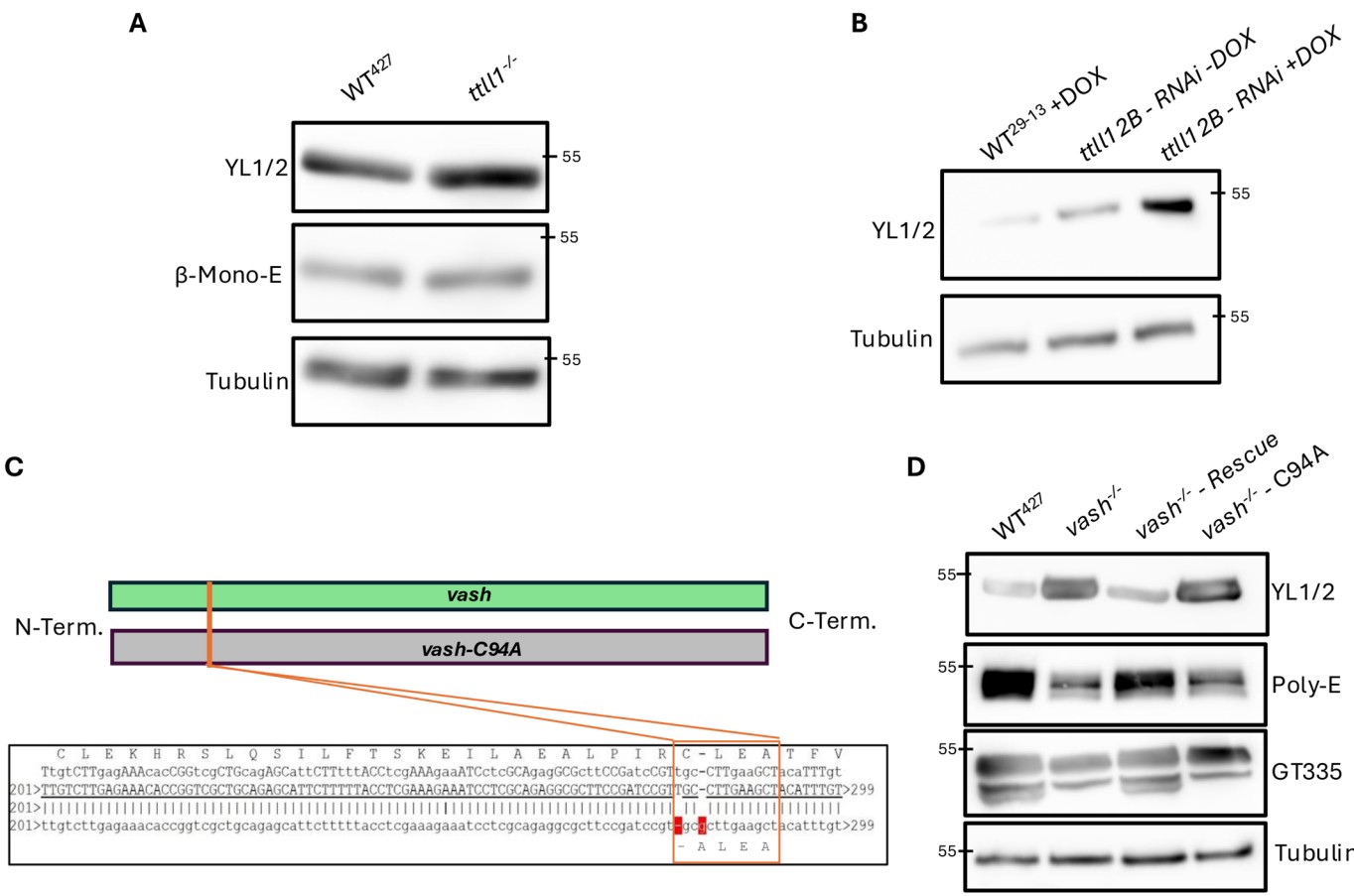

**Fig. 5. Microtubule polyglutamylation and detyrosination regulate each other.** (A) Western blot analysis of tyrosination levels and monoglutamylation levels in cytoskeletal extracts of WT[427] and ttll1[−/−] cells. (B) Western blot analysis of tyrosination levels in cytoskeletal extracts of wild-type (WT[29-13]), and ttll12B - RNAi cells. The wild-type cells and the ttll12B - RNAi cells were incubated for 3 days with doxycycline prior to analysis, to induce the RNAi, while additionally the ttll12b - RNAi cells without doxycycline were also analysed. (C) Schematic depiction of the creation of a vasohibin-dead variant (vash-C94A). Two single point mutations were introduced to change the cysteine 94 in the enzymatic active CLEA motive to alanine 94. (D) Western blot analysis of tyrosination and polyglutamylation levels in cytoskeletal extracts of WT[427], vash[−/−], vash[−/−] - Rescue and vash[−/−] -C94A cells. The 55 kDa band of the used PageRuler Protein Ladder is indicated on all blots as a molecular weight orientation.

vash[−/−] - C94A were indistinguishable, confirming the catalytic deficiency of this dead mutant (Fig. 5D). Additionally, we found the same changes in polyglutamylation levels compared to the knockout cell line (Fig. 5D). The Poly-E signal was significantly lower compared to that of the wild-type and rescue, and the band pattern of the GT335 western blot was identical to that of the knockout cells, indicating that this cross-regulation between polyglutamylation and detyrosination was not based on the presence of VASH serving as a potential platform for TTLL recruitment and activation. Rather, the catalytic activity of VASH is essential for its stimulatory effect on polyglutamylation.

## DISCUSSION

The term 'tubulin code', proposed by Verhey and Gaertig (2007), refers to the expression of various α- and β-tubulin isotypes in combination with the presence of a multitude of PTMs, such as polyglutamylation, polyglycylation and acetylation. However, the first tubulin PTM discovered was the detyrosination/tyrosination cycle of the C-terminal tyrosine of α-tubulin (Barra et al., 1973). In the present study, we address two research questions. First, what is the role of tyrosination and detyrosination of the microtubule cytoskeleton of the protozoan parasite *T. brucei*, and, second, is there any cross-talk between the tyrosination/detyrosination status and polyglutamylation?

We used gene deletion, gene rescue and overexpression of the two enzymes catalysing detyrosination and tyrosination to study their function. First, we confirmed results from a previous study (van der Laan et al., 2019) that TbVash is the only tubulin carboxypeptidase in *T. brucei*. Deletion of both alleles abolished the presence of detectable levels of detyrosinated tubulin. This affected both α- and β-tubulin, because, in contrast to other organisms, *T. brucei* has β-tubulin with a C-terminal tyrosine that is a substrate for TbVASH (Schneider et al., 1997). In *vash*[−/−] cells, the axoneme flagellum is also highly tyrosinated. In wild-type cells, tyrosinated axonemal microtubules are only transiently observed in the growing, newly formed flagellum. The mature axoneme is completely detyrosinated. Artificially induced tyrosination of the axoneme and the absence of structural abnormalities confirms that detyrosination is not a prerequisite of stable microtubule strictures, but a consequence (Webster et al., 1990).

To study possible effects of the absence of translation-independent tubulin tyrosination, we deleted *T. brucei TTL*. Because the tyrosinated tubulin pool is constantly replenished by *de-novo* translation, *TTL* deletion will only partially deplete microtubules of tyrosinated tubulin. To further enhance detyrosination, we overexpressed *TbVASH* in the *ttl*[−/−] cells. This led indeed to a further significant reduction in tyrosinated microtubules.

In a previous study, it was shown that a CRISPR-Cas9-mediated gene inactivation of *TbVASH* led to an impaired growth phenotype associated with mitotic defects and abnormally elongated cells (van der Laan et al., 2019). However, in our study, using conventional gene deletion by homologous recombination, we did not observe these phenotypes. In mammalian cells, detyrosination of spindle microtubules contribute to the regulation of microtubule-kinetochore attachment (Girao et al., 2024). However, in *T. brucei*, the protein composition of kinetochores is fundamentally different from that observed in other model organisms (Akiyoshi and Gull, 2014). This could explain the refractiveness of mitosis in *T. brucei* against changes in detyrosination levels. Instead, the population growth curve and generation time were not significantly different from those of the parental wild-type cells. Also, we did not observe any major morphological alterations, both at light and electron microscopy level. However, subtle but significant effects on the length and shape were detectable in *vash*^{−/−} cells. These effects were specific because they were rescuable by ectopic re-expression of *VASH*. A recent study in the related intracellular parasite *Leishmania mexicana*, also a kinetoplastid, reported the deletion of the *LmVASH* homolog (Corrales et al., 2025). Here, and more compatible with our results, the growth phenotype and morphological alterations were relatively mild. However, these parasites were significantly impaired in infecting mice. A possible explanation is the observation that the rudimentary flagellum, present in the parasite life cycle stage and responsible for spreading an infection from cell to cell, displayed structural defects in *vash*^{−/−} cells, such as a further shortening and possible defects in intraflagellar transport. However, in the flagellum of the *T. brucei* mutants, we did not observe deviations in length or ultrastructure in comparison to wild-type cells.

Despite the lack of significant morphological and growth-related phenotypes, a careful analysis of the motility of the *vash*^{−/−} hypertyrosinated cells revealed a significant variation of the motility characteristics. As compared to wild-type cells, the *vash*^{−/−} cells show a pronounced shift to remaining longer in the run state, in which cells swim in a directed fashion, with a comparable velocity to that of the wild-type cells. The statistics of tumble periods, during which cells do not show an appreciable directed movement (i.e. the counterpart or complement of the run state), were highly similar in all cell lines. As a consequence, *vash*^{−/−} cells featured a roughly 30% increase in the run phase fraction. In contrast, hypotyrosinated *ttl*^{−/−} - Vash^{OE} cells showed motility characteristics that were very similar to those of wild-type cells. In the algae *Chlamydomonas reinhardtii*, a differential tyrosination/detyrosination pattern was observed for A- and B-tubules of the flagellar axonemes (Johnson, 1998). Disturbing this balance in *VASH* knockout algae led to aberrant intraflagellar transport (IFT) organisation (Chhatre et al., 2025). However, such a differential pattern is not observed in *T. brucei*. Instead, axonemal microtubules of the flagellum are quantitively detyrosinated (see Fig. 1D). Nevertheless, detyrosination might be important for IFT processes and *vash*^{−/−}-induced tyrosination of the axoneme might impact IFT and therefore flagellar assembly and cell motility. It should, however, be noted that the motility phenotype is not necessarily a direct consequence of axonemal hypertyrosination. In *vash*^{−/−} cells, reduction in glutamylation of the axoneme is also observed due to cross-talk between these PTMs (see Fig. S5B and Discussion below), and therefore motility defects could be the result of a combination of PTM defects. It has been shown that both tyrosination and polyglutamylation affect the activity of motor proteins, such as kinesins and dynein (McKenney et al., 2016; Suryavanshi et al., 2010).

Because trypanosomes have evolved sophisticated patterns of motility that are essential for colonisation and survival in the hosts and vector (Broadhead et al., 2006; Engstler et al., 2007; Rotureau et al., 2014), we hypothesise that subtle changes in mechanisms of motility induced by an imbalanced microtubule PTMs might have an impact on the virulence of the parasite, similar to the observation made in *Leishmania* (Corrales et al., 2025). However, this remains to be investigated in an animal model of infection.

Microtubule detyrosination, polyglutamylation and polyglycylation occur within the C-terminal ten to 15 amino acids of both α- and β-tubulin (Westermann and Weber, 2003). Therefore, it is not unreasonable to hypothesise cross-talk and possible functional dependencies between these PTMs. Since *T. brucei* lacks polyglycylation, it presents a simplified system to address this question.

Data from two recent studies supported this potential relationship between polyglutamylation and tyrosination levels. Using synthetic peptides based on the C-terminal sequence of mammalian α-tubulin, Mahalingan et al. (2020) found that peptides without the C-terminal tyrosine and terminating with glutamate served as a much better substrate for recombinant mouse TTLL6, a polyglutamylase that adds long glutamate side chains to the C-terminal tail of α-tubulin. Using semisynthetic tubulin dimers, engineered to contain a defined C-terminal α-tubulin tail by fusing a synthetic peptide to a truncated recombinant α-tubulin protein, Ebberink et al. (2023) showed that a polyglutamylated tail promoted detyrosination in a manner correlating with the length of the glutamyl side chain. This apparent cross-talk between the detyrosination status and polyglutamylation was corroborated by recent observation in *Leishmania in vivo*, showing that a *VASH* gene knockout, causing increased tyrosination, led to a decrease in polyglutamylation (Corrales et al., 2025). Also, it has been shown that tubulin re-tyrosination contributes to the control of tubulin acetylation (Martinez-Hernandez et al., 2022; Xu et al., 2017). However, this is most likely an indirect effect, because acetylation stabilises microtubules and therefore results in an increase in polyglutamylation.

Having generated both *VASH* and *TTL* and, in our previous studies (Jentzsch et al., 2020, 2024), polyglutamylase (*TTLL*) knockout trypanosome cells, we investigated this potential cross-talk in more detail. We show that the rate of polyglutamylation is inversely correlated to tyrosination levels. A high level of tyrosination is associated with a decrease in polyglutamylation, in particular a decrease in long side chains, while low levels of tyrosinated tubulin showed the reverse correlation. To note, all PTM measurements derive from antibody-based readouts of adjacent C-terminal epitopes, which could, in principle, influence one another's accessibility. However, consistent trends across GT335, Poly-E, and β-mono-E labelling, together with genetic rescue (*vash*^{−/−} - *Rescue*) and the catalytically inactive variant (*vash*^{−/−} - *C94A*), reduce the likelihood that a single antibody artifact explains the observed effects. Our observations are also consistent with previous *in vitro* data using synthetic peptides mimicking the tubulin C-termini (Ebberink et al., 2023; Mahalingan et al., 2020). Also, currently available antibodies against tubulin PTMs are not exhaustive. In *T. brucei*, using mass spectrometry, it has recently been shown that not only the canonical E435 site of β-tubulin is polyglutamylated, but also the E438 site (Nisavic et al., 2025). No specific antibody is currently available to include this modification site in a functional analysis and therefore we cannot assess any cross-talk of detyrosination/ryrosination with this additional polyglutamylation site on β-tubulin. Using a selection of TTLL-deficient cell lines, we were able to show that this relationship

appears to form a positive feedback loop: detyrosination stimulates polyglutamylation, which, in turn, stimulates further detyrosination. This interconnection might be mostly based on a cross-talk between detyrosination and the elongation of the glutamyl side chains, since the Poly-E antibody signal in the $vash^{-/-}$ was mainly affected while the branching point signal (β-mono-E) was not. Cells depleted of the polyglutamylases TTLL1 and TTLL12B, which we previously identified as elongases, showed the biggest impact on detyrosination (Jentzsch et al., 2020, 2024). Furthermore, this relationship is not based on a protein-protein interaction between VASH and the TTLLs, since analysis using an enzymatically inactive $vash^{-/-}$ mutant did not display this PTM cross-talk. Therefore, this cross-regulation is dependent on the presence of the specific microtubule modifications.

## MATERIALS AND METHODS
### Cell culture
Procyclic *T. brucei* Lister strain 427 cells were cultivated at 27°C in SDM-79 medium (Life Technologies, UK), which was supplemented with additional 10% fetal bovine serum (Capricorn Scientific, Germany) and 7.5 mg/l hemin (Merck, Germany). The knockout cell lines were selected wit 1 µg/l puromycin and 10 µg/l blasticidin. The rescue cell lines, and the *VASH* expression cell line were additionally selected with 50 µg/l hygromycin. For the growth curve analysis, a CASY cell counter (Roche Innovatis AG, Germany) was used. The data for the growth curves were the averages of three biological replicates, each consisting of three technical replicates. The averages and standard deviations are listed in Table S2. RNAi cell line *TTLL12B* and *TTLL1* knockout cell line are described in Jentzsch et al. (2020) and Jentzsch et al. (2024), respectively.

### Generation of *VASH and TTL* gene deletions
Modified versions of the pBlueScript vector with either a puromycin or a blasticidin resistance gene were used for the generation of the gene knockouts. Approximately 150 bp of the 5′ and 3′ untranslated regions of the target genes were amplified and integrated into the vector using the SbfI/PacI (insertion of the 5′ homology region upstream of the resistance gene) and the FseI/AscI (insertion of the 3′ homology region downstream of the resistance gene) restriction sites. For the transfection the cassette containing the 5′ homology region, the respective resistance gene and the 3′ homology region was cut out from the vector using the SbfI/AscI restriction sites. This linear DNA fragment was then used for the transfection into the cells by electroporation utilising an Amaxa Nucleofactor II (Lonza, Germany) (Burkard et al., 2007). Screening for clones with successful integrated resistance cassettes was done by PCR. Therefore, the genomic DNA was isolated by phenol/chloroform extraction. The primer pairs were designed with one primer binding outside of the target DNA sequence to correctly verify an accurate integration event. A correct double knockout had both wild-type alleles replaced with their respective resistance cassettes (Fig. S1A-C) The resulting gene deletion strains are *Δvash::PURO/Δvash::BLA*, referred to as $vash^{-/-}$, and *Δttl::PURO/Δttl::BLA*, referred to as $ttl^{-/-}$. All oligonucleotides are listed in Table S3.

### Ectopic expression of *VASH* and *vash*-C94A
The open reading frame of the *VASH* was PCR amplified and integrated into the promoterless pTag8-tub vector using the FseI/AscI restriction sites (Wickstead et al., 2003). The vector was then linearised through the NotI restriction site and used for transfection into the knockout cell lines $vash^{-/-}$ and $ttl^{-/-}$, where it integrates into the tubulin locus (Wickstead et al., 2003). The resulting cell lines are *Δvash::PURO/Δvash::BLA/vash(exp.)*, referred to as $vash^{-/-}$ - Rescue, and *Δttl::PURO/Δttl::BLA/vash(exp.)*, referred to as $ttl^{-/-}$ - Vash$^{OE}$. The same expression vector was used for $vash^{-/-}$ - Rescue and *VASH* overexpression in the $ttl^{-/-}$ - Vash$^{OE}$ cell line. This is because the overexpression is expression in addition to the endogenous gene expression, whereas rescue is expression in the absence of endogenous *VASH* genes. For the mutation of cysteine 94 to alanine in the *VASH* open reading frame, three overlapping fragments were produced, based on the prior constructed

pTag8-tub vector containing the *VASH* open reading frame. The first fragment was the vector backbone cut at the XhoI/KpnI restriction sites. The second fragment overlapped with the first fragment at the XhoI restriction site and reaches up to the mutation site, changing the codon from TGC to GCG. The third fragment overlaps with the second one at the mutation site, also including the same mutation, and with the vector backbone at the KpnI restriction site. Fragments were assembled using the NEBuilder HiFi DNA Assembly kit (New England Biolabs, Germany). The following transfection into the $vash^{-/-}$ knockout cell line was performed as described above. The resulting cell line is *Δvash::PURO/Δvash::BLA/vash-C94A(exp.)*, referred to as $vash^{-/-}$ - C94A. All oligonucleotides are listed in Table S3.

### Endogenous tagging of *T. brucei VASH*
The C-terminal endogenous tagging of Vash with 3xTy-mNeonGreen was performed as described by Dean et al. (2015) using the pPOTv7-phleomycin-3xTy-mNG-3xTy vector. The resulting cell line is *VASH/vash-3xTy-mNG*, referred to as Vash-3xTy-mNG. The primer sequences were obtained from the TrypTag database (Billington et al., 2023). The Ty tag is an epitope tag recognised by the monoclonal mouse antibody BB2 (Bastin et al., 1996).

### Reverse transcription qPCR
To validate transcript levels of *VASH* and *TTL* in the knockout and expression cell lines, their RNA was isolated using the RNeasy Plus Mini Kit (Qiagen, Germany) and transcribed into cDNA with the RevertAid First Strand cDNA Synthesis Kit (Thermo Fisher Scientific). The qPCR was performed using 5 ng of cDNA template with Maxima SYBR Green/ROX qPCR Master Mix (Thermo Fisher Scientific) and a StepOne real-time PCR system (Thermo Fisher Scientific). As endogenous control, the constitutively expressed gene *PFR-A* was used, and the gene expression levels were then calculated using the ΔΔCT method. All oligonucleotides used are listed in Table S3. The qPCR reaction protocol is stated in Table S4.

### Microscopy
In general, cells were settled on poly-L-lysin-coated microscopy slides and washed once with PBS, for all protocols. For the preparation of the cytoskeleton, the settled cells were prepared as previously described (Schock et al., 2021). Isolated flagella were prepared from the settled cells by incubation with 100 µl of 1% NP40 in PEME (2 mM EGTA, 1 mM MgSO$_4$, 0.1 mM EDTA, 0.1 M piperazine-*N*,*N*′-bis(2-ethanesulfonic acid)–NaOH, pH 6.9) for 5 min, before being incubated thrice with 100 µl 1% NP40 and 1 M NaCl in PEME for 5 min each (Robinson et al., 1991). Regardless, the cells/ cytoskeletons/flagella were then fixated through dehydration in −20°C ice-cold methanol for at least 30 min, before being rehydrated with PBS. Then, they were incubated for 1 h with the primary antibody in a wet chamber, washed thrice with PBS, then incubated for another hour in the wet chamber with the secondary antibody. Finally, they were washed once with PBS, then incubated with DAPI (1 µg/ml in PBS) and then washed with PBS again, before being mounted with Vectashield Antifade Mounting Medium (Vector Labs).

Depending on the primary antibody, some deviations from the described protocols were made. For the KMX staining the cells were settled on the slides first, before being fixated chemically by 3.5% formaldehyde, 5% acetic acid and 0.1% NP40 for 15 min. For the Kinesin 13-1 staining, the cells were chemically fixated in 4% formaldehyde for 15 min before being washed twice with PBS and then being settled on the microscopy slides. Then the cells were permeabilised with 0.1% NP40 in PEME for 20 min, washed once with PBS and then the slides were blocked for 30 min with blocking buffer [1% blocking reagent (Roche) in 100 mM maleic acid, 150 mM sodium chloride, pH 7.5]. The primary and secondary antibody were each diluted in blocking buffer, before being used. Fluorescent *in situ* hybridisation and ultrastructure expansion microscopy was performed as described (Ersfeld and Gull, 1997; Kalichava and Ochsenreiter, 2021). The Zeiss Axio Imager M2 microscope equipped with a PCO Panda 4.2 sCMOS camera (Visitron Systems, Germany) and operated with the VisiView software (Visitron Systems) was used for all microscopies. Image acquisition and processing for motility analysis was done as described

(Jentzsch et al., 2020, 2024). Details of the motility analysis are described in the Results section and in more technical detail in Jentzsch et al. (2024).

## SDS-PAGE sample preparation and western blotting
Samples for SDS-PAGE of cytoskeletons and flagellar fractions were prepared as previously described (Jentzsch et al., 2024). SDS-PAGE, optimised to separate tubulin monomers with β-tubulin running further than α-tubulin, was conducted as described (Banerjee et al., 2010). Western blots were done according to standard methods and probed with primary and HRP-conjugated secondary antibodies. The blots were developed by chemiluminescence using the Lumigen substrate (Takara, Japan) and an ImageQuat LAS-400 detection system (GE Healthcare). Comparative western blots were quantified using the tubulin loading control as normalisation standard (Fig. S6).

## Flow cytometry
For the flow cytometric analysis of the cell cycle distribution, $6 \times 10^6$ cells from a mid-log cell culture were harvested, washed once with PBS and resuspended in 250 μl 0.5% formaldehyde. The suspension was then placed on ice for 5 min, before 2.5 ml of −20°C ice-cold 70% ethanol was added dropwise while vortexing. The solution was then rotated for 1 h at 4°C, and the cells were pelleted, resuspended in 1 ml of staining solution (50 μg/ml propidium-iodide, 20 μg/ml RNAseA in PBS) and incubated for 30 min at 37°C. Prior to analysis the cell suspension was diluted 1:10 in PBS. The analysis was performed with the Cytomics FC 500 (Beckman Coulter, Krefeld) and the associated CXP software.

## Image quantification and statistical analysis
Image analysis, overlays and measurements were conducted with ImageJ version 1.53 (Schneider et al., 2012). Deconvolution of images was performed by the ImageJ plugin 'Parallel Spectral Deconvolution 2D' using the Generalized Tokhonov (reflexive) method (https://imagej.net/plugins/parallel-spectral-deconvolution, 28 August 2024). The applied point-spread-functions were prior calculated by the 'Diffraction PSF 3D' plugin (https://imagej.net/plugins/diffraction-psf-3d, 28 August 2024). All morphometric analysis was conducted on interphase cells, identifiable by containing one non-dividing nucleus and one kinetoplast. Cell and flagellum length measurements with the free-hand drawn line measurement tool from ImageJ, with the line centred in the middle of the cell, ranging from the posterior to the anterior pole. Measurements of tip width, width at the kinetoplast and width at the nucleus were performed using the Image J line measurement tool as shown in Fig. S3. 'Roundness' of the posterior cell tip was quantified using the ImageJ Measure function (Fig. S7). To measure the fluorescence intensity difference between anterior and posterior pole, a circular area of fixed and equal size was chosen and placed near the respective poles. Statistical analysis was performed using GraphPad Prism 8 Version 8.0.2 (263) (GraphPad Software, Inc.). The Mann–Whitney $U$ test was used for pairwise comparison, while the Kruskal–Wallis test was used for comparison between multiple samples. All figures were assembled in Microsoft PowerPoint.

## Antibodies
The following primary antibodies were used: mouse monoclonal IgG anti-α-tubulin [TAT1 (Woods et al., 1989); gift from Keith Gull, University of Oxford, UK]; rat monoclonal IgG anti-tyrosinated α-tubulin (YL1/2, Sigma-Aldrich, Cat. No. MAB1864); rabbit polyclonal anti-detyrosinated α-tubulin and rabbit polyclonal anti-detyrosinated β-tubulin (gifts from Krzysztof Rogowski, Université de Montpellier, France); sheep anti-digoxigenin Fab fragment (anti-DIG, Boehringer Mannheim); mouse monoclonal anti-β-tubulin (KMX, gift from Keith Gull); rabbit polyclonal anti-Kinesin 13-1 (Chan et al., 2010); mouse monoclonal anti-myc (clone 9E10, Developmental Studies Hybridoma Bank, Iowa, USA); mouse monoclonal anti-polyglutamylated α/β-tubulin (GT335, Biomol, Cat. No. AG-20B-0020-C100, detects short and long side chains); rabbit polyclonal anti-polyglutamylated α/β-tubulin (Poly-E, Biomol, Cat. No. AG-25B-0030-C050, detects long side chains); rabbit polyclonal anti-monoglutamylated β-tubulin (β-mono-E; gift from Carsten Janke, Institute Curie, University Paris-Saclay, France). A review of antibodies directed against tubulin posttranslational modifications, including those used in this

study, is given by Magiera and Janke (2013); mouse monoclonal anti-Ty epitope (BB2, gift from Keith Gull) (Bastin et al., 1996). The following secondary antibodies were used: anti-mouse IgG–HRP conjugate (Sigma-Aldrich, Cat. No. A9044); anti-rat IgG-HRP conjugate (Invitrogen, Cat. No. 31460), anti-rabbit IgG–HRP conjugate (Biozol, Cat. No. IMU-DKXRB-003-DHRPX); anti-mouse IgG-Atto-488 conjugate (Sigma-Aldrich, Cat. No. 41698); anti-mouse IgG-Atto-550 conjugate (Sigma-Aldrich, Cat. No. 43394); anti-rabbit IgG–CF488A conjugate (Sigma-Aldrich, Cat. No. SAB4600234); anti-rabbit IgG-Alexa-Fluor-555 conjugate (Invitrogen, Cat. No. A-21428); anti-rat IgG–Alexa-Fluor-488 conjugate (Invitrogen, Cat. No. A-11006); and anti-sheep IgG-FITC conjugate (The Jackson Laboratory, Cat. No. 313-095-045).

## Acknowledgements
We thank Keith Gull (University of Oxford, UK) for the TAT1, KMX and BB2 antibodies; Carsten Janke (Institute Curie, University Paris-Saclay, France) for the β-mono-E antibody; Krzysztof Rogowski (Université de Montpellier, France) for the anti-detyrosinated α- and β-tubulin antibodies; and Samuel Dean (University of Warwick, UK) for the pTAG-tub and pPOTv7-phleomycin-3xTy-mNG plasmids.

## Competing interests
The authors declare no competing or financial interests.

## Author contributions
Conceptualization: M.W., K.E.; Data curation: M.T., H.W., K.E.; Formal analysis: M.T., H.W., L.B., M.W., K.E.; Funding acquisition: M.W., K.E.; Investigation: M.T., H.W., L.B., S.W., K.E.; Methodology: M.T., H.W., L.B., S.W., M.W., K.E.; Project administration: M.W., K.E.; Resources: M.W., K.E.; Software: H.W., M.W.; Supervision: M.W., K.E.; Validation: M.T., H.W., M.W., K.E.; Visualization: M.T., H.W., M.W.; Writing – original draft: H.W., K.E.; Writing – review & editing: K.E.

## Funding
Financial support by the Deutsche Forschungsgemeinschaft priority program 'Physics of Parasitism' (ER692/3-1, ER692/3-2, WE4335/5-1 and WE4335/5-2) is gratefully acknowledged. Open Access funding provided by Universität Bayreuth. Deposited in PMC for immediate release.

## Data and resource availability
All relevant data and details of resources can be found within the article and its supplementary information.

## First Person
This article has an associated First Person interview with the first author of the paper.

## Peer review history
The peer review history is available online at https://journals.biologists.com/bio/lookup/doi/10.1242/bio.062270.reviewer-comments.pdf

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
