## [Peer Review File · Biology Open]

Effects of microtubule (de)tyrosination on the morphology and motility of *Trypanosoma brucei* and crosstalk with polyglutamylation

Marinus Thein, Hannes Wunderlich, Lucas Brehm, Stella Wagner, Matthias Weiss and Klaus Ersfeld

DOI: 10.1242/bio.062270

Editor: Sandhya Koushika

Review timeline

Submission to sister journal:	7 July 2025
Editorial decision at sister journal:	28 August 2025
Transfer to Biology Open:	25 September 2025
Editorial decision:	25 October 2025
First revision received:	5 November 2025
Accepted:	7 November 2025

Original submission to sister journal

First decision letter

MS TITLE: Effects of microtubule (de)tyrosination and its crosstalk with polyglutamylation on the morphology and motility of *Trypanosoma brucei* and crosstalk with polyglutamylation

AUTHORS: Marinus Thein; Hannes Wunderlich; Lucas Brehm; Stella Wagner; Matthias Weiss; Klaus Ersfeld

ARTICLE TYPE: Research Article

It has taken longer than I would like because of the difficulty of finding reviewers during the summer, but we have now reached a decision on the above manuscript. To see the reviewers' reports and a copy of this decision letter, please go to:

As you will see from their reports, the reviewers have very different opinions concerning your study. After careful consideration and reading your paper again I am afraid that I will not be inviting revision. I am very sorry to give you such disappointing news, but it takes a very enthusiastic recommendation by the referees for a manuscript to progress further.

I do hope you find the comments of the reviewers helpful in allowing you to revise the manuscript for submission elsewhere, and many thanks for sending your work to

Comments from the Reviewers:

Reviewer 1: SUMMARY OF THE ADVANCE MADE IN THIS PAPER AND ITS POTENTIAL SIGNIFICANCE TO THE FIELD

In this manuscript, Marinus Thein et al. reported that microtubule tyrosination affects trypanosome cell motility and cross-talks with tubulin polyglutamylation. Post-translational modifications of microtubules in trypanosomes are poorly explored, and recent work from Dr. Klaus Ersfeld's lab has started to uncover the roles of tubulin modification in regulating cytoskeleton integrity and cell motility. However, whether tubulin tyrosination and polyglutamylation are interdependent and whether there is any feedback control between tubulin post-translational modifications in

trypanosomes remain elusive, although the interdependency and feedback control have been reported in other systems previously. In the current work, the authors examined, first, the effect of tubulin tyrosination on cell viability and cell motility in the procyclic form of *T. brucei*, and they found that modulation of tubulin tyrosination affected cell motility without impacting cell viability. The authors further analyzed the interplay between tubulin tyrosination and polyglutamylation and found the interdependency between them. Overall, the work presented in this manuscript represents incremental advances in the field and has moderate impact in the understanding of tubulin post-translational modifications in trypanosome biology.

SUGGESTIONS TO AUTHORS

Major comments:

1. The importance of tubulin tyrosination/detyrosination in trypanosome biology appears to be limited. Although it affects cell motility, there is little impact on cell proliferation. The authors implied that it may be involved in host infection, but this was not explored in this study. Therefore, the presented work has limited impact in the field.
2. The interdependency between tubulin tyrosination and glutamylation has been observed in other systems previously. The discovery of similar control mechanism in trypanosomes suggests that it is an evolutionarily conserved process, which makes such findings in trypanosomes less impactful.

Minor comments:

1. There is no quantitation of the western blots, which is critical to evaluate the changes in tubulin modifications in control, RNAi and rescue cells.
2. In Figure 5D, there is no decrease of GT335 in *vash-/-C94A* cell line, but the text stated there was a decrease in abundance.

Reviewer 2: SUMMARY OF THE ADVANCE MADE IN THIS PAPER AND ITS POTENTIAL SIGNIFICANCE TO THE FIELD

SYNOPSIS:

The authors examine impact of altering tubulin PTMs, detyrosination and polyglutamylation in the protozoan, *Trypanosoma brucei*. They use gene knockout, combined with biochemistry, immunofluorescence, morphometric and motility analyses to assess the impact of KO of genes for adding (TTL) or removing (VASH) C-terminal tyrosine from tubulin. This is an important question, both for the trypanosome research community and for the broader community studying roles of tubulin PTM. The work is rigorous, presented clearly and convincingly, with novel and important findings.

Two particularly notable results are:

1. The authors find expansion of tubulin tyrosination on microtubules, including on the axoneme, in the VASH knockout alters the run and tumble pattern of trypanosomes, causing cells to run for longer periods. This is relevant for transmission and pathogenesis as it could conceivably influence the ability of the parasite to navigate through host tissues, because alteration in run/tumble periods can be used by organisms to mediate chemotaxis.
2. The authors observe cross-talk between detyrosination and polyglutamylation PTMs in *T. brucei* cells, with one PTM influencing the prevalence of the other. While cross talk has been suggested from in vitro studies, demonstrating this in live cells is, to my knowledge, novel.

Notably loss of either TTL or VASH did not substantially affect viability or cell doubling time, though did cause cell morphology defects. I expect the work will be of interest to the JCS readership. I have only a few comments and suggestions to offer to the authors to consider for strengthening the work.

SUGGESTIONS TO AUTHORS

Major comments [Please request additional experiments only if they are essential for supporting the conclusions; authors should be encouraged to highlight any claims that are preliminary or speculative, or to discuss any pitfalls or alternative interpretations in a 'Limitations' section]

Minor comments

COMMENTS

*  Can the authors use a morphometric analysis to better assess the morphological defect in the KOs? The VASH defect in particular, likely will be revealed with more clarity if for example they could quantitatively re-report roundness of posterior end.

*  lines 214-216. Please provide the citation for the prior CRISPR VASH KO study noted.

*  please use motile instead of "mobile" when motile is meant (a moveable object is "mobile" but not "motile")

*  The analysis of explanation for greater fraction of cells in the run period (lines 238 - 275) is overly complicated. It seems to me that if a greater percentage of cells are in the run phase, the only explanation is that they run for longer. If they "switched more frequently" between run and tumble, without spending more time running, then the fraction of cells in the run phase would be unchanged vs control.

*  The discussion is a bit long and could be shortened without negatively impacting the work.

*  The title has redundancies to be removed.

Transfer to Biology Open

Author response to reviewers' comments

Comments to the suggestions by the reviewers

Reviewer 1:

It is correct that other, previous studies have suggested a crosstalk between microtubule deetyrosination and polyglutamylolation. However, in all of these studies this observation was either anecdotal and never systematically investigated or was obtained by in vitro experiments. As far as we are aware, our study is the first to focus on this interdependency and investigate this in vivo and therefore we disagree that our data are of limited impact.

A quantification of all western blots has now been included in a new Supplementary figure. It is correct that the intensity of the GT335 western blot signal in the Vash-/-C94A line is similar in intensity to wild-type, but please note the complete absence of the lowest band on this western, which is also absent in the vash-/- cells, but present in wild-type and Vash rescue cells. However, the difference is more obvious on the Poly-E blot. This is more relevant anyway, as only poly-E detects long glutamyl side chains. We have changed the text accordingly.

Reviewer 2:

We have now included a (basic) morphometric analysis using an ImageJ plug in to measure "roundness" of an object. These data are in a new Supplementary figure. The term "mobile" has

been replaced with “motile”.

We agree with the reviewer that our statement on being more often or being longer in the run phase may have read somewhat redundant and potentially confusing. However, the mere fraction only is a sufficiently informative quantity if the distributions of periods spent in the run and tumble phases are not heavy-tailed, switching between the two complementary states is Markovian, and tumble phases are not shortened. Since all these aspects are initially unknown, one needs to inspect the two distributions individually. We therefore believe that our approach is not overly complicated but rather a necessary step to be able to draw sound conclusions. To clarify our approach in easy terms, we have rephrased the sentence in lines 239-242.

We have noticed that the title in the document returned to the reviewers is not our original title. This must have been an error that occurred during manuscript processing.

First decision letter

MS ID#: bio.062270

MS TITLE: Effects of microtubule (de)tyrosination and its crosstalk with polyglutamylatonon the morphology and motility of *Trypanosoma brucei* and crosstalk with polyglutamylation

AUTHORS: Marinus Thein; Hannes Wunderlich; Lucas Brehm; Stella Wagner; Matthias Weiss; Klaus Ersfeld

ARTICLE TYPE: Research Article

I have now reached a decision on the above manuscript.

The reviewer reports are shown at the bottom of this email or can be accessed, together with a copy of this decision letter, by going to:

As you will see, the reviewers gave favourable reports, but raised some critical points that will require amendments to your manuscript. I hope that you will be able to carry these out, because we would like to be able to accept your paper.

At this stage, we also ask you to ensure your manuscript complies with our formatting guidelines - please see our manuscript preparation guidelines for details. Provided you are able to fully address the referees' comments, we are positive about publication of your paper (we accept over 95% of revision submissions) and therefore hope you won't mind any extra work involved in reformatting your manuscript at this point.

Please upload both a 'clean' version of your Word file, along with a highlighted version clearly showing where you have made changes in the revised manuscript. Please avoid using 'Track changes' in Word files as these are lost in PDF conversion.

I should be grateful if you would also provide a point-by-point response detailing how you have dealt with the points raised by the reviewers in the 'Response to Reviewers' box. Please attend to all of the reviewers' comments. If you do not agree with any of their criticisms or suggestions please explain clearly why this is so.

Reviewer 1

Comments for the author

The tubulin code is emerging as one of the key regulators of the microtubule cytoskeleton across species. While we are still understanding the nuances of how each of the posttranslational

modifications play key roles in regulating specific aspects of cell and organismal function, one key aspect that is emerging across studies is the potential cross talk between the diverse modifications.

In their current study, Thein et al have approached this intriguing aspect of how two specific posttranslational modifications, tyrosination/detyrosination and glutamylation/deglutamylation can influence the flagella-induced motility of the intracellular parasite *Trypanosoma brucei*. Using genetic models of the organism lacking either the tyrosinase or the detyrosinase, they have tried to delineate the role of tubulin tyrosination cycle in influencing cellular motility. They first establish specific KO lines of both VASH and TTL and show that these cells show distinct differences in their levels of these modifications, while not having a great deal of impact on cell cycle progression and growth parameters. They then go on to assess the effect of altered tyrosination/detyrosination on cell motility and see increase in the run phase in cells lacking VASH, which was mainly due to a prolonged run period due to reduced detyrosination. They conclude that the levels of detyrosination control whether the cells are in the running or tumbling phase. Finally, to understand if this is influenced by a crosstalk between tyrosination/detyrosination and polyglutamylation, the authors have tested the levels of polyglutamylation in both the VASH KO and the TTL KO lines and see that in the VASH KO lines, there is a marked decrease in polyglutamylation.

While the manuscript is well written, experiments succinctly planned and outlined, there is no mechanistic link as to whether the reduction in detyrosination is coupled to increased polyglutamylation also in the flagella and if this could influence the changes in cell motility. Moreover, the fact that these changes do not influence the cell cycle is surprising, considering gene inactivation has been shown to affect mitosis before. The authors mention it and do not discuss in detail. Hence, at this stage, the manuscript is describing a known phenomenon that has not been looked into in this model organism, which is okay, but this needs to be strengthened further by addressing the points I raise below:

Major points:

1. Compared to reports in mammalian cells, the authors do not see any defects in the mitotic spindle arrangement and cell cycle distribution in their VASH KO line. What is the reason behind this? Is there a separate mechanism that overcomes this, considering lack of detyrosination can lead to MT detachment from the kinetochores (Girão H et al., 2024. Nat Commun)
2. Is the impact of the loss of VASH on the cell length observed in Fig. 2A also due to the delay in the progression of the cells through the different mitotic phases?
3. In Fig. 3A, the authors only comment on motility of TTL-/-VASH OE cells but not the TTL-/- cells. Did they not quantify it or were they totally immobile? This is an interesting point of data and it would be good to include it as part of the study.
4. Recent work in *Chlamydomonas* shows that loss of VASH leads to IFT defects and shorter flagella (Chhatre A et al., 2025. Nat Commun). Do the authors observe shorter flagella in their cells as well? This is not shown or discussed in the manuscript and needs to be included. If they do not see any change, this needs to be discussed as to why they think this could be the case.
5. Does the cell size influence the overall motility of the cells? Considering smaller cells may have shorter flagella, does that impact the cell motility?
6. Do the flagella of the vash-/- cells also have reduced glutamylation as they have increased tyrosination? The authors need to show the levels of glutamylation in the flagella more than just in the antero-posterior poles as the levels of glutamylation within the flagella would influence how the axonemal dynein function, thus regulating their motility.
7. In their discussion, the authors claim that their data shows that rate of glutamylation depends on tyrosination levels. How can they comment on the rate of glutamylation through immunostaining and western blots? They just show that the levels are modulated. The authors need to tone this down as assessing the rate of modification will require more experimental work with purified proteins and in vitro analyses.

8. The authors need to discuss more on how this crosstalk is influencing cell motility, highlighting the potential reasons behind the changes in motility they observe. The discussion in its current state is just stating facts rather than giving potential hypotheses towards understanding the overall regulation of flagellar motility by the fine-tuning of these two tubulin modifications.

Minor points

1. It would be nice to have a graphical representation of the localisation of the tyrosinated-vs-detyrosinated tubulin would be observed in the WT cells, for a better understanding of the changes in the overall distribution of the signal in the immunofluorescence data presented in Fig. 1C and Fig. 2A.

2. Since the localisation of the YL1/2 signal is more confined to the basal body and kinetoplasts, as has been shown earlier as well, does it mean that in the normal cells, the rest of the MTs including the flagella are detyrosinated and hence, more stable? This understanding of the distribution would help better appreciate the impact of this localised arrangement of a specific subset of MTs.

3. Page 10, line 228 typographical error: established methodology (see Jentzsch et al (2024) for details).

4. Page 11, line 289: This is factually incorrect that GT335 binds to glutamyl chains shorter than 3 residues. It is actually an antibody that binds to the branch point glutamate, which can thus recognise both the short chain and long chain glutamates (Wolff et al., 1992). However, the specificity of the antibody reduces as the length of the glutamate chain becomes longer. The authors need to correct this statement accordingly.

5. In many places, the reference to the figures have been placed incorrect. For example, in page 11, line 293: ... the specific polyglutamylation branching at E435 on b-tubulin (Fig. 4 A and S5A).... refers to the use of the b-mono-E antibody, while the figure 4A is actually talking about the use of PolyE and GT335. So, I recommend that the figure numbers are mentioned where the text refers to it accordingly.

6. The signal for GT335 in the vash-/- cells are quite varied in the panels 4A and S5A. The authors need to have panels that represent similar signals for the cells in different blots.

7. The figure panel 4B is very difficult to comprehend in terms of the figure nomenclature. It is better to have an easier and simpler way of representation.

8. Figure 4D and 5D, the blot for GT335 is incomprehensible due to overload of the protein. A better blot panel is needed.

9. None of the western blot panels have molecular weight markers indicated. Moreover, the staining for antibodies like PolyE and GT335 show double bands. What do these bands correspond to? There is no clarity on this in the figure. The authors need to amend them with corrected panels.

10. Supplementary videos of cells in motility would be good to see the phenotype apart from only looking at the quantifications. It will make for a better understanding of what the authors are claiming in terms of motility defects.

Reviewer 2

Comments for the author

This manuscript presents an elegant and timely study exploring the crosstalk between tubulin detyrosination and polyglutamylation in *Trypanosoma brucei*. The authors propose that these two post-translational modifications (PTMs) engage in a positive feedback loop, whereby detyrosination enhances polyglutamylation and polyglutamylation promotes detyrosination. They further show that excessive tyrosination leads to subtle morphological defects and pronounced motility changes

(prolonged run phases without affecting viability). The work is of interest to the broader cytoskeleton and parasite cell biology communities.

Major comments

Evidence for a feedback loop.

The conclusion that detyrosination and polyglutamylation form a bidirectional feedback loop relies entirely on immunolabelling with antibodies recognizing adjacent C-terminal epitopes (YL1/2, GT335, and Poly-E). Because these epitopes are in close proximity, alterations in one modification could directly affect the accessibility of the neighboring epitopes, potentially leading to reciprocal detection artifacts. In this scenario, changes in antibody signal might reflect steric or conformational effects rather than true biochemical feedback. Without controls distinguishing these possibilities, the apparent reciprocity between Figures 4 and 5 could be partly methodological.

The authors could clarify this in the text by emphasizing convergent evidence from multiple antibodies and genetic rescues, for example "All PTM measurements derive from antibody-based readouts of adjacent C-terminal epitopes, which could, in principle, influence one another's accessibility. However, consistent trends across GT335, Poly-E, and β -mono-E labelling, together with genetic rescue ($vash^{-/-}$ -Rescue) and the catalytically inactive variant ($vash^{-/-}$ -C94A), reduce the likelihood that a single antibody artifact explains the observed effects."

YL1/2 antibody specificity.

YL1/2 recognizes tyrosinated α -tubulin but also binds RP2, a transition-fiber protein at basal bodies (Andre et al., 2014, J. Biol. Chem.), which may explain residual YL1/2 signal in the $ttl^{-/-}$ and $ttl^{-/-}$ - $vashOE$ mutants (Figures 1A, 1C). Because of this cross-reactivity, relative quantification across Western blots should be interpreted cautiously, even though the overall trends remain convincing.

The manuscript should also acknowledge broader limitations of antibody detection, including (i) YL1/2 detecting only α -tubulin tyrosination, not β -tubulin, and (ii) potential steric hindrance among adjacent epitopes as noted above.

Soluble versus polymerized tubulin pools.

Most experiments are performed on detergent-extracted cytoskeletons or flagella. It would be valuable for the authors to comment on whether the detected PTMs represent modifications on polymerized microtubules only or whether soluble tubulin pools could also contribute. Both tyrosination and glutamylation can occur on soluble tubulin dimers as well as on polymers. A brief discussion would strengthen interpretation of the antibody-based assays.

Attribution of motility phenotypes.

The motility phenotype is attributed to hyper-tyrosination, but the corresponding mutants also display reduced polyglutamylation. Since both PTMs independently influence microtubule-based motility (tyrosination modulating motor initiation (McKenney et al., 2016, EMBO J), and polyglutamylation tuning dynein activity (Suryavanshi et al., 2010, Curr Biol)). These concurrent changes could confound phenotype interpretation. The authors should discuss this interdependence and clarify that motility alterations may reflect combined rather than isolated PTM effects.

Site specificity of β -tubulin glutamylation.

Recent proteomic evidence shows that β -tubulin E438 is also glutamylated in *T. brucei* (Nisavic et al., 2025, J. Proteome Res). Since the β -mono-E antibody specifically recognizes the E435 site, the authors should comment on whether their conclusions might be affected by this site specificity.

Statistical significance

The authors should provide more detailed information on the sample sizes and number of biological and technical replicates used for each statistical analysis, to allow proper assessment of the robustness and reproducibility of the results

Minor comments

Line 132: Trypanosome brucei → Trypanosoma brucei (T. brucei)

The motility section is technically dense; adding a short conceptual summary before introducing equations (for example "We next assessed whether tubulin modification affects cell motility dynamics using a trajectory analysis method (Jentzsch et al., 2024).") would aid readability.

The crosstalk section is elegant but long; consider subheadings.

Figure 1E: orient the legend vertically within the graph for readability.

Figure 2A: specify how cell body and flagellum lengths were measured (in the figure legend or in the materials and methods).

Figure 2B: the right panel could be moved to Supplementary Data to avoid overemphasizing rare elongated cells.

While the absence of TbVASH expression in the $vash^{-/-}$ cell line was confirmed by RT-PCR, the authors should also verify the double knockout at the genomic level. In addition to confirming deletion of both alleles, it is important to rule out the possibility that residual TbVASH expression, or relocalization of the ORF elsewhere in the genome, could account for the unexpectedly mild phenotype compared with van der Laan et al. (2019). This is particularly relevant because detection of VASH by antibody relies on a single tagged allele, which would not reveal any untagged protein. Such verification would strengthen confidence that the observed phenotypes are solely due to complete loss of TbVASH.

Figure 4B: label WT and $vash^{-/-}$ cells directly on the IFA image for readability.

Line 500: remove duplication of "gene."

Clarify why identical pTAg8b-Tub vectors yield "Rescue" versus "Overexpression" designations if expression levels are comparable.

The TAT antibody should be cited as TAT1 (Woods et al., 1989, J. Cell Sci.) as another TAT antibody is commercialized and is not specific to tubulin.

This is a well-conceived and potentially impactful study that contributes meaningfully to understanding the tubulin code in a divergent eukaryote. The main conclusions are well supported but would benefit from a more explicit discussion of the limitations inherent in antibody-based PTM detection. With these clarifications, the paper will make a valuable contribution to the field.

Reviewer's Responses to Questions

Experimental quality

Does each figure have the proper controls?

If 'No', please indicate reasons in Comments for Author box below.

Reviewer #1:

- Yes

Reviewer #2:

- Yes

Were the data analyzed using appropriate statistical tests?

If 'No', please indicate reasons in Comments for Author box below.

Reviewer #1:

- Yes

Reviewer #2:

- Yes
-

Reproducibility

Were experiments performed using adequate number of biological replicates?

If 'No', please indicate reasons in Comments for Author box below.

Reviewer #1:

- Yes

Reviewer #2:

- Yes
-

Does the methods section provide sufficient detail to permit reproducibility?

If 'No', please indicate reasons in Comments for Author box below.

Reviewer #1:

- Yes

Reviewer #2:

- Yes
-

Completeness

Are the manuscript's conclusions supported by the data?

If 'No', please indicate reasons in Comments for Author box below.

Reviewer #1:

- No

Reviewer #2:

- Yes
-

Scholarship

Do the authors cite and discuss the merits of data that would argue for and against their conclusion?

If 'No', please indicate reasons in Comments for Author box below.

Reviewer #1:

- Yes

Reviewer #2:

- Yes
-

Does the manuscript title & abstract accurately reflect the contents of the manuscript, without hyperbole?

If 'No', please indicate reasons in Comments for Author box below.

Reviewer #1:

- Yes

Reviewer #2:

- Yes
-

First revision

Author response to reviewers' comments

We have tried to address all comments returned to us by the two reviewers. All changes in the manuscript are written in red letters and addressed point-by-point below. I would like to point out that we found the comments by the reviewers to be very constructive and we think this has contributed considerably to improve the manuscript.

Comments from the Reviewers:

Reviewer 1: The tubulin code is emerging as one of the key regulators of the microtubule cytoskeleton across species. While we are still understanding the nuances of how each of the posttranslational modifications play key roles in regulating specific aspects of cell and organismal function, one key aspect that is emerging across studies is the potential cross talk between the diverse modifications.

In their current study, Thein et al have approached this intriguing aspect of how two specific posttranslational modifications, tyrosination/detyrosination and glutamylation/deglutamylation can influence the flagella-induced motility of the intracellular parasite *Trypanosoma brucei*. Using genetic models of the organism lacking either the tyrosinase or the detyrosinase, they have tried to delineate the role of tubulin tyrosination cycle in influencing cellular motility. They first establish specific KO lines of both VASH and TTL and show that these cells show distinct differences in their levels of these modifications, while not having a great deal of impact on cell cycle progression and growth parameters. They then go on to assess the effect of altered tyrosination/detyrosination on cell motility and see increase in the run phase in cells lacking VASH, which was mainly due to a prolonged run period due to reduced detyrosination. They conclude that the levels of detyrosination control whether the cells are in the running or tumbling phase. Finally, to understand if this is influenced by a crosstalk between tyrosination/detyrosination and polyglutamylation, the authors have tested the levels of polyglutamylation in both the VASH KO and the TTL KO lines and see that in the VASH KO lines, there is a marked decrease in polyglutamylation.

While the manuscript is well written, experiments succinctly planned and outlined, there is no mechanistic link as to whether the reduction in detyrosination is coupled to increased polyglutamylation also in the flagella and if this could influence the changes in cell motility. Moreover, the fact that these changes do not influence the cell cycle is surprising, considering gene inactivation has been shown to affect mitosis before. The authors mention it and do not discuss in detail. Hence, at this stage, the manuscript is describing a known phenomenon that has not been looked into in this model organism, which is okay, but this needs to be strengthened further by addressing the points I raise below:

Major points:

1. Compared to reports in mammalian cells, the authors do not see any defects in the mitotic spindle arrangement and cell cycle distribution in their VASH KO line. What is the reason behind this? Is there a separate mechanism that overcomes this, considering lack of detyrosination can lead to MT detachment from the kinetochores (Girão H et al., 2024. Nat Commun)

We have included a possible explanation for this difference in the Discussion

2. Is the impact of the loss of VASH on the cell length observed in Fig. 2A also due to the delay in the progression of the cells through the different mitotic phases?

As the generation time is similar between wild-type and mutants, this is unlikely. We have included this in Results

3. In Fig. 3A, the authors only comment on motility of TTL-/-VASH OE cells but not the TTL-/- cells. Did they not quantify it or were they totally immobile? This is an interesting point of data and it would be good to include it as part of the study.

The motility of TTL^{-/-} cells was indistinguishable from wild-type cells. We have included this statement in Results

4. Recent work in Chlamydomonas shows that loss of VASH leads to IFT defects and shorter flagella (Chhatre A et al., 2025. Nat Commun). Do the authors observe shorter flagella in their cells as well? This is not shown or discussed in the manuscript and needs to be included. If they do not see any change, this needs to be discussed as to why they think this could be the case.

We did not observe shorter flagella and have included this observation in the Results

5. Does the cell size influence the overall motility of the cells? Considering smaller cells may have shorter flagella, does that impact the cell motility?

TTL^{-/-OE} cells are also smaller than wild-type, but have no motility defect. This argues against an effect of cell size on motility (with a certain range, of course)

6. Do the flagella of the vash^{-/-} cells also have reduced glutamylation as they have increased tyrosination? The authors need to show the levels of glutamylation in the flagella more than just in the antero-posterior poles as the levels of glutamylation within the flagella would influence how the axonemal dynein function, thus regulating their motility.

Yes, they have and we have included this point in Results and Discussion

7. In their discussion, the authors claim that their data shows that rate of glutamylation depends on tyrosination levels. How can they comment on the rate of glutamylation through immunostaining and western blots? They just show that the levels are modulated. The authors need to tone this down as assessing the rate of modification will require more experimental work with purified proteins and in vitro analyses.

Yes, we agree that a functional connection is not proven, just a correlation. We have modified our statements accordingly and added references to previous in vitro studies showing a similar connection.

8. The authors need to discuss more on how this crosstalk is influencing cell motility, highlighting the potential reasons behind the changes in motility they observe. The discussion in its current state is just stating facts rather than giving potential hypotheses towards understanding the overall regulation of flagellar motility by the fine-tuning of these two tubulin modifications.

Currently, this is only an observation of a correlation between these two PTMs. We and others, who have made similar observation, do not know anything about the mechanics behind this connection. We have at least addressed that it is not the mere presence of Vash that might be important for recruiting polyglutamylases, but that it is the polyglutamylation itself. Any hypothesis to explain this correlation is currently only highly speculative and therefore we refrained from presenting a model for this phenomenon.

Minor points

1. It would be nice to have a graphical representation of the localisation of the tyrosinated-vs-detyrosinated tubulin would be observed in the WT cells, for a better understanding of the changes in the overall distribution of the signal in the immunofluorescence data presented in Fig. 1C and Fig. 2A.

We have decided not to do this because it would oversimplify the situation.

2. Since the localisation of the YL1/2 signal is more confined to the basal body and kinetoplasts, as has been shown earlier as well, does it mean that in the normal cells, the rest of the MTs including the flagella are detyrosinated and hence, more stable? This understanding of the distribution would help better appreciate the impact of this localised arrangement of a specific subset of MTs.

The staining of the basal body with YL1/2 is probably a cross-reactivity with a non-tubulin protein of the basal body (RP2). We were made aware of this possibility by the second reviewer and have included this in Results. It is correct that in wild-type cells the axoneme and most of the sub-pellicular cytoskeleton, except the posterior end, are completely detyrosinated. This has been shown in other publications as well. Overall, the cytoskeleton is extremely stable and not dynamic. We have now mentioned this in the Introduction. References of publications describing the cytoskeleton of trypanosomes were already included in the manuscript.

3. Page 10, line 228 typographical error: established methodology (see Jentzsch et al (2024) for details).

Corrected.

4. Page 11, line 289: This is factually incorrect that GT335 binds to glutamyl chains shorter than 3 residues. It is actually an antibody that binds to the branch point glutamate, which can thus recognise both the short chain and long chain glutamates (Wolff et al., 1992). However, the specificity of the antibody reduces as the length of the glutamate chain becomes longer. The authors need to correct this statement accordingly.

Corrected.

5. In many places, the reference to the figures have been placed incorrect. For example, in page 11, line 293: ... the specific polyglutamylation branching at E435 on β -tubulin (Fig. 4 A and S5A).... refers to the use of the β -mono-E antibody, while the figure 4A is actually talking about the use of PolyE and GT335. So, I recommend that the figure numbers are mentioned where the text refers to it accordingly.

Corrected

6. The signal for GT335 in the vash-/- cells are quite varied in the panels 4A and S5A. The authors need to have panels that represent similar signals for the cells in different blots.

A better panel has been inserted in 4A.

7. The figure panel 4B is very difficult to comprehend in terms of the figure nomenclature. It is better to have an easier and simpler way of representation.

We have re-arranged the labelling of Fig. 4B and hope this is now easier to comprehend

8. Figure 4D and 5D, the blot for GT335 is incomprehensible due to overload of the protein. A better blot panel is needed.

The GT335 blot in 5D has been replaced. In 4D the GT335 blot has been included for completeness. There are no changes between the different cell types and therefore it has no consequences for our interpretation of the data.

9. None of the western blot panels have molecular weight markers indicated. Moreover, the staining for antibodies like PolyE and GT335 show double bands. What do these bands correspond to? There is no clarity on this in the figure. The authors need to amend them with corrected panels.

This has now been added and a statement has been included to clarify which band is alpha- and beta-tubulin.

10. Supplementary videos of cells in motility would be good to see the phenotype apart from only looking at the quantifications. It will make for a better understanding of what the authors are claiming in terms of motility defects.

The analysis of motility was done using hundreds of individual cell tracks from many videos. Since the motility defects are subtle they are difficult to spot by looking at individual cells. Therefore, we did not show any individual videos.

Reviewer 2: This manuscript presents an elegant and timely study exploring the crosstalk between tubulin detyrosination and polyglutamylation in *Trypanosoma brucei*. The authors propose that these two post-translational modifications (PTMs) engage in a positive feedback loop, whereby detyrosination enhances polyglutamylation and polyglutamylation promotes detyrosination. They further show that excessive tyrosination leads to subtle morphological defects and pronounced motility changes (prolonged run phases without affecting viability). The work is of interest to the broader cytoskeleton and parasite cell biology communities.

Major comments

Evidence for a feedback loop.

The conclusion that detyrosination and polyglutamylation form a bidirectional feedback loop relies entirely on immunolabelling with antibodies recognizing adjacent C-terminal epitopes (YL1/2,

GT335, and Poly-E). Because these epitopes are in close proximity, alterations in one modification could directly affect the accessibility of the neighboring epitopes, potentially leading to reciprocal detection artifacts. In this scenario, changes in antibody signal might reflect steric or conformational effects rather than true biochemical feedback. Without controls distinguishing these possibilities, the apparent reciprocity between Figures 4 and 5 could be partly methodological.

The authors could clarify this in the text by emphasizing convergent evidence from multiple antibodies and genetic rescues, for example "All PTM measurements derive from antibody-based readouts of adjacent C-terminal epitopes, which could, in principle, influence one another's accessibility. However, consistent trends across GT335, Poly-E, and β -mono-E labelling, together with genetic rescue ($vash^{-/-}$ -Rescue) and the catalytically inactive variant ($vash^{-/-}$ -C94A), reduce the likelihood that a single antibody artifact explains the observed effects."

That is a very good suggestions and we have adapted and included this statement.

YL1/2 antibody specificity.

YL1/2 recognizes tyrosinated α -tubulin but also binds RP2, a transition-fiber protein at basal bodies (Andre et al., 2014, J. Biol. Chem.), which may explain residual YL1/2 signal in the $ttl^{-/-}$ and $ttl^{-/-}$ - $vashOE$ mutants (Figures 1A, 1C). Because of this cross-reactivity, relative quantification across Western blots should be interpreted cautiously, even though the overall trends remain convincing.

The manuscript should also acknowledge broader limitations of antibody detection, including (i) YL1/2 detecting only α -tubulin tyrosination, not β -tubulin, and (ii) potential steric hindrance among adjacent epitopes as noted above.

We were not aware of this possibility as the information is a bit hidden in the Andre-paper. This is of course a very important information which we have now included.

Soluble versus polymerized tubulin pools.

Most experiments are performed on detergent-extracted cytoskeletons or flagella. It would be valuable for the authors to comment on whether the detected PTMs represent modifications on polymerized microtubules only or whether soluble tubulin pools could also contribute. Both tyrosination and glutamylation can occur on soluble tubulin dimers as well as on polymers. A brief discussion would strengthen interpretation of the antibody-based assays.

This is a hypothetical possibility. Glutamylation can occur on soluble tubulin, but as far as I know this has only been shown using in vitro assays. Tubulin tyrosination can only happen on soluble dimers, but not on polymers. However, the soluble tubulin pool in trypanosomes is very small indeed. We didn't find any literature to cite, but from our own experiments we estimate it to be less than 1% of total tubulin and is unlikely to be of relevance for our findings. We don't want to add this to the manuscript, but below is a small blot showing this. Insoluble and soluble fraction were blotted and probed with an anti-tubulin antibody. In order to get a non-saturated signal in the insoluble (cytoskeleton) fraction, the samples use for these lanes were diluted 1:20. The soluble sample is undiluted. Both bloodstream (BSF) and procyclic (PCF) form trypanosomes were analysed:

Attribution of motility phenotypes.

The motility phenotype is attributed to hyper-tyrosination, but the corresponding mutants also display reduced polyglutamylation. Since both PTMs independently influence microtubule-based motility (tyrosination modulating motor initiation (McKenney et al., 2016, EMBO J), and polyglutamylation tuning dynein activity (Suryavanshi et al., 2010, Curr Biol)). These concurrent changes could confound phenotype interpretation. The authors should discuss this interdependence and clarify that motility alterations may reflect combined rather than isolated PTM effects.

Again, a very valuable comment and we have included this in the Discussion

Site specificity of β -tubulin glutamylation.

Recent proteomic evidence shows that β -tubulin E438 is also glutamylated in *T. brucei* (Nisavic et al., 2025, J. Proteome Res). Since the β -mono-E antibody specifically recognizes the E435 site, the authors should comment on whether their conclusions might be affected by this site specificity.

We have included this information in our Discussion.

Statistical significance

The authors should provide more detailed information on the sample sizes and number of biological and technical replicates used for each statistical analysis, to allow proper assessment of the robustness and reproducibility of the results

The information on statistical significance (n, tests) is actually included in each figure legend and in Material and Methods.

Minor comments

Line 132: Trypanosome brucei → Trypanosoma brucei (*T. brucei*)

Corrected.

The motility section is technically dense; adding a short conceptual summary before introducing equations (for example "We next assessed whether tubulin modification affects cell motility dynamics using a trajectory analysis method (Jentzsch et al., 2024).") would aid readability.

Yes, good suggestions and we have now included this.

The crosstalk section is elegant but long; consider subheadings.

Done

Figure 1E: orient the legend vertically within the graph for readability.

I guess the reviewer refers to the colour code of the lines. We have rearranged the legend.

Figure 2A: specify how cell body and flagellum lengths were measured (in the figure legend or in the materials and methods).

This information has been added to Materials and Methods.

Figure 2B: the right panel could be moved to Supplementary Data to avoid overemphasizing rare elongated cells.

We discuss this in the text and emphasize the rare abundance of longer cells. We would prefer to keep the statistics in Fig. 2.

While the absence of TbVASH expression in the *vash^{-/-}* cell line was confirmed by RT-PCR, the authors should also verify the double knockout at the genomic level. In addition to confirming deletion of both alleles, it is important to rule out the possibility that residual TbVASH expression, or relocalization of the ORF elsewhere in the genome, could account for the unexpectedly mild phenotype compared with van der Laan et al. (2019). This is particularly relevant because detection of VASH by antibody relies on a single tagged allele, which would not reveal any untagged protein. Such verification would strengthen confidence that the observed phenotypes are solely due to complete loss of TbVASH.

We have included the PCR data on genomic DNA as new Supplementary Fig. 1B

Figure 4B: label WT and *vash^{-/-}* cells directly on the IFA image for readability.

We have marked the cells with arrows of different colour.

Line 500: remove duplication of "gene."

Corrected.

Clarify why identical pTA_{g8b}-Tub vectors yield "Rescue" versus "Overexpression" designations if expression levels are comparable.

An explanation has been added to Materials and Methods.

The TAT antibody should be cited as TAT1 (Woods et al., 1989, J. Cell Sci.) as another TAT antibody is commercialized and is not specific to tubulin.

We have changed this accordingly and added the reference

This is a well-conceived and potentially impactful study that contributes meaningfully to understanding the tubulin code in a divergent eukaryote. The main conclusions are well supported but would benefit from a more explicit discussion of the limitations inherent in antibody-based PTM detection. With these clarifications, the paper will make a valuable contribution to the field.

Second decision letter

MS ID#: bio.062270R1

MS TITLE: Effects of microtubule (de)tyrosination and its crosstalk with polyglutamylatonon the morphology and motility of *Trypanosoma brucei* and crosstalk with polyglutamylatonon

AUTHORS: Marinus Thein; Hannes Wunderlich; Lucas Brehm; Stella Wagner; Matthias Weiss; Klaus Ersfeld

ARTICLE TYPE: Research Article

I am happy to tell you that your manuscript has been accepted for publication in Biology Open, pending our standard publication integrity checks. It was accepted on 7th November 2025.